# Maternal immune response and placental antibody transfer after COVID-19 vaccination across trimester and platforms

Caroline G. Atyeo [1,2,10], Lydia L. Shook [3,4,10], Sara Brigida [4], Rose M. De Guzman [3,4], Stepan Demidkin[4], Cordelia Muir[4], Babatunde Akinwunmi [5], Arantxa Medina Baez[4], Maegan L. Sheehan[1], Erin McSweeney[4], Madeleine D. Burns [3,6,7,8], Ruhi Nayak[4], Maya K. Kumar[5], Chinmay D. Patel[5], Allison Fialkowski [4,6], Dana Cvrk[3], Ilona T. Goldfarb [3], Lael M. Yonker [6,7,8], Alessio Fasano [6,7,8], Alejandro B. Balazs[1], Michal A. Elovitz [9], Kathryn J. Gray[5,11], Galit Alter [1,11✉] & Andrea G. Edlow [3,4,11✉]

The availability of three COVID-19 vaccines in the United States provides an unprecedented opportunity to examine how vaccine platforms and timing of vaccination in pregnancy impact maternal and neonatal immunity. Here, we characterize the antibody profile after Ad26.COV2.S, mRNA-1273 or BNT162b2 vaccination in 158 pregnant individuals and evaluate transplacental antibody transfer by profiling maternal and umbilical cord blood in 175 maternal-neonatal dyads. These analyses reveal lower vaccine-induced functions and Fc receptor-binding after Ad26.COV2.S compared to mRNA vaccination and subtle advantages in titer and function with mRNA-1273 versus BN162b2. mRNA vaccines have higher titers and functions against SARS-CoV-2 variants of concern. First and third trimester vaccination results in enhanced maternal antibody-dependent NK-cell activation, cellular and neutrophil phagocytosis, and complement deposition relative to second trimester. Higher transplacental transfer ratios following first and second trimester vaccination may reflect placental compensation for waning maternal titers. These results provide novel insight into the impact of platform and trimester of vaccination on maternal humoral immune response and transplacental antibody transfer.

[1] Ragon Institute of MGH, MIT, and Harvard, Cambridge, MA, USA. [2] PhD Program in Virology, Division of Medical Sciences, Harvard University, Boston, MA, USA. [3] Department of Obstetrics & Gynecology, Massachusetts General Hospital, Harvard Medical School, Boston, MA, USA. [4] Vincent Center for Reproductive Biology, Massachusetts General Hospital, Boston, MA, USA. [5] Department of Obstetrics and Gynecology, Brigham and Women's Hospital, Harvard Medical School, Boston, MA, USA. [6] Mucosal Immunology and Biology Research Center, Massachusetts General Hospital, Boston, MA, USA. [7] Department of Pediatrics, Massachusetts General Hospital, Boston, MA, USA. [8] Harvard Medical School, Boston, MA, USA. [9] Maternal and Child Health Research Program, Department of Obstetrics and Gynecology, University of Pennsylvania, Philadelphia, PA, USA. [10]These authors contributed equally: Caroline G. Atyeo, Lydia L. Shook. [11]These authors jointly supervised this work: Kathryn J. Gray, Galit Alter, Andrea G. Edlow. ✉email: GALTER@MGH. HARVARD.EDU; AEDLOW@MGH.HARVARD.EDU

Pregnant individuals with COVID-19 are not only at increased risk for severe morbidity and mortality[1–4], but also for adverse pregnancy outcomes including preterm delivery, pregnancy loss, and stillbirth[5–10]. While vaccination against COVID-19 is a critically important public health strategy to protect pregnant individuals and their pregnancies, approximately one-third of pregnant individuals in the U.S. remain unvaccinated, with the majority (57.7%) vaccinated prior to pregnancy and only ~10% opting for vaccination during pregnancy, according to the U.S. Centers for Disease Control and Prevention's most recent statistics[11]. Because pregnant individuals were excluded from initial vaccine clinical trials[12–14], data to guide clinical decision-making in this population have lagged behind those for the general population, contributing to vaccine hesitancy. To date, studies have demonstrated that pregnant people mount robust immunological responses to COVID-19 mRNA vaccines (BNT162b2 and mRNA-1273) with final titers achieved being comparable to those in non-pregnant women of reproductive age[15–17], and with similar safety and reactogenicity profiles[15,17–19]. Population-level data have demonstrated the effectiveness of COVID-19 vaccines in protecting pregnant people from severe/critical COVID-19 and maternal mortality[9,20–24]. Several studies of pregnant people receiving COVID-19 mRNA vaccines primarily in the third trimester have also demonstrated the presence of anti-SARS-CoV-2-specific antibodies capable of neutralization and immune effector functions in umbilical cord blood at delivery[15–17,25–28], and recent data from the CDC demonstrate that maternal mRNA vaccination is 61% effective in preventing newborn hospitalization from COVID-19 in the first 6 months of life[29]. Maternal vaccination against COVID-19 thus has the potential not only to protect the pregnant individual, but to confer fetal and neonatal benefits by preventing adverse pregnancy outcomes related to severe maternal COVID-19 illness, and by providing newborns with immunity through transplacental and breastmilk transfer of maternal antibodies[30,31].

Little is known, however, regarding how trimester-specific pregnancy immunity and different COVID-19 vaccine platforms may interact to impact maternal and neonatal protection from COVID-19. Trimester-specific immunological adaptations occur during normal pregnancy to promote implantation, support fetal growth and development, and stimulate parturition[32–35]. Although COVID-19 vaccine safety across all trimesters of pregnancy has been well demonstrated[18,19,36–38], whether the trimester of vaccination impacts vaccine immunogenicity or transplacental transfer to the neonate remains incompletely understood, as many pregnancies in which vaccination occurred in the first and early second trimester were ongoing at the time of initial study publications[15–17,25–27]. Furthermore, studies comparing immune responses across COVID-19 vaccine platforms are limited even in the non-pregnant population[39–41], and platform-specific immune responses in the pregnant population have been limited to one comparison of mRNA-1273 vs BNT162b2[16]. To date, no studies have directly assessed the immunogenicity of the Ad26.CoV.2 vaccine in pregnancy, nor compared these profiles to mRNA vaccines across gestation. To address these gaps, we used an unbiased systems serology approach to characterize the maternal antibody response and transplacental antibody transfer to the umbilical cord by vaccine platform (BNT162b2, mRNA-1273, or Ad26.CoV2) and by trimester of vaccination.

## Results

**Clinical and demographic information**. Using systems serology, we profiled the vaccine-induced immune response in a cohort of 158 women who completed a COVID-19 vaccine course during pregnancy: 28 who received Ad26.COV2.S (1 dose), 61 who received mRNA-1273 (2 doses), and 69 who received BNT162b2 (2 doses). Cohort demographics by vaccine platform are shown in Table 1. There was no significant difference in the number of days elapsed from the second dose of mRNA vaccines or the single dose of Ad26.COV2.S vaccine to time of participant sample (median days [IQR]: 62 [27–91], 42 [22–74], and 58 [38–85] for mRNA-1273, BNT162b2 and Ad26.COV2.S respectively, $p = 0.10$). There were no differences between study groups (vaccine platforms) in maternal age, gravidity, parity, pre-pregnancy BMI, race, insurance status, or presence of an autoimmune disorder. Individuals who received the Ad26.COV2.S vaccine were more likely to be of Hispanic ethnicity.

We initially evaluated transplacental antibody transfer via systems serology for those participants who had delivered at the time of maternal antibody profiling ($n = 123$ maternal-neonatal dyads, Supplemental Table 1). There were no differences in gestational age at delivery, mode of delivery, neonatal sex, or neonatal birthweight by vaccine platform. To enhance understanding of transplacental antibody transfer by trimester of vaccination, IgG titers against Spike were quantified using ELISA in these 123 dyads and an additional 52 dyads who had delivered by study completion. In this set of 175 dyads, 27 participants (15%) were vaccinated with Ad26.COV2.S, 62 (35%) with mRNA-1273, and 86 (49%) with BNT162b2. Supplemental Table 2 depicts vaccine type and days elapsed from second dose (or single dose if receiving Ad26.COV2.S) to delivery by trimester of vaccination for this expanded dyad cohort.

**Maternal vaccine immune response by vaccine platform**. To begin to understand differences in the vaccine-induced immune response across the three vaccine platforms, we plotted the Spike-specific antibody titer and Fc-receptor (FcR) binding in maternal serum (Fig. 1A and Fig. S1A). These plots reveal that whereas similar antibody profiles against Spike were observed for the two mRNA vaccines (mRNA-1273 and BNT162b2), vaccine-induced antibody titers and FcR-binding across all IgG subclasses and two antibody isotypes (IgG and IgA) were significantly lower in individuals who received the Ad26.COV2.S vaccine. Overall, the anti-Spike response was similar in individuals who received the mRNA vaccines mRNA-1273 and BNT162b2, with the exception of a significantly higher IgG2 anti-Spike response in women vaccinated with mRNA-1273 compared to those vaccinated with BNT162b2 (Fig. S1A). Individuals who received Ad26.COV2.S also displayed significantly lower antibody functions (Fig. 1B and Fig. S1C), as measured by antibody-dependent cellular phagocytosis (ADCP), antibody-dependent neutrophil phagocytosis (ADNP), antibody-dependent complement deposition (ADCD) and antibody-dependent NK-cell activation (ADNKA, measured as % CD107a+, % MIP-1β+ and % IFNγ+ cells). Antibody titer and FcR-binding against the Spikes from variants of concern (Alpha, Beta, Delta, and Gamma) were highly correlated with response to the ancestral Spike, suggesting that individuals who mount a robust vaccine-induced antibody response will have antibodies against variants of concern (Fig. S1B). While the overall reduction in antibody effector functions observed in Ad26.CoV2.S recipients is likely of greatest clinical relevance regardless of titer, to elucidate whether reduced antibody effector functions noted in Ad26.COV2.S recipients were simply due to lower antibody titer, we adjusted the functional measurements by IgG1 titer. This analysis confirmed that mRNA vaccines induced higher ADCP, ADNP, and ADNKA activity compared to the Ad26.COV2.S vaccine, independent of IgG1 titer (Fig. S2A). Differences in ADCD effector functions were no longer significant after adjusting for titer.

**Table 1 Demographic and clinical characteristics of maternal participants by vaccine platform.**

| | Overall (N = 158) | Ad26.COV2.S (N = 28) | mRNA-1273 (N = 61) | BNT162b2 (N = 69) | P |
|---|---|---|---|---|---|
| Maternal age, years | 34 [32, 36] | 34 [32, 36] | 33 [32, 36] | 34 [32, 36] | 0.78 |
| Gravidity | 2 [1, 3] | 2 [2, 3] | 2 [1, 3] | 2 [1, 3] | 0.27 |
| Parity | 1 [0, 1] | 1 [0, 1] | 1 [0, 1] | 1 [0, 1] | 0.73 |
| Race (%) | | | | | |
|  Asian | 9 (6) | 1 (4) | 3 (5) | 5 (7) | 0.36 |
|  Black or African American | 4 (3) | 2 (7) | 1 (2) | 1 (1) | |
|  Other | 6 (4) | 3 (11) | 2 (3) | 1 (1) | |
|  Unknown/not reported | 3 (2) | 1 (4) | 1 (2) | 1 (1) | |
|  White | 135 (85) | 21 (75) | 54 (89) | 60 (87) | |
|  American Indian or Alaskan native | 1 (1) | 0 (0) | 0 (0) | 1 (1) | |
| Ethnicity (%) | | | | | |
|  Hispanic | 9 (6) | 4 (14) | 3 (5) | 2 (3) | 0.02 |
|  non-Hispanic | 141 (89) | 20 (71) | 57 (93) | 64 (93) | |
|  Unknown/not reported | 8 (5) | 4 (14) | 1 (2) | 3 (4) | |
| Insurance status (%) | | | | | |
|  Private | 152 (96) | 25 (89) | 61 (100) | 66 (96) | 0.05 |
|  Public | 5 (3) | 3 (11) | 0 (0) | 2 (3) | |
|  Unknown | 1 (1) | 0 (0) | 0 (0) | 1 (1) | |
| Pre-pregnancy BMI, kg/m$^2$ | 24 [22, 27] | 23 [22, 27] | 23 [22, 27] | 24 [22, 26] | 1.00 |
| Obesity (%) | | | | | |
|  No | 118 (75) | 22 (79) | 42 (69) | 54 (78) | 0.47 |
|  Yes | 40 (25) | 6 (21) | 19 (31) | 15 (22) | |
| Autoimmune condition (%) | | | | | |
|  No | 152 (96) | 28 (100) | 58 (95) | 66 (96) | 0.74 |
|  Yes | 6 (4) | 0 (0) | 3 (5) | 3 (4) | |
| Prior infection with SARS-CoV-2[a] (%) | | | | | |
|  No | 153 (97) | 26 (93) | 58 (95) | 69 (100) | 0.06 |
|  Yes | 5 (3) | 2 (7) | 3 (5) | 0 (0) | |
| Trimester of vaccination (%) | | | | | |
|  First | 18 (11) | 2 (7) | 7 (11) | 9 (13) | 0.05 |
|  Second | 88 (56) | 13 (46) | 42 (69) | 33 (48) | |
|  Third | 52 (33) | 13 (46) | 12 (20) | 27 (39) | |
| Time from vaccination to sample collection[b], days | 50 [27, 86] | 58 [38, 85] | 62 [27, 91] | 42 [22, 74] | 0.10 |

*BMI* body mass index.
[a]Four participants tested positive for SARS-CoV-2 prior to vaccination and one participant tested positive at delivery.
[b]Defined time from a single dose of Ad26.COV2.S vaccine or second dose of mRNA-1273 or BNT162b2 and sample collection. Continuous variables presented as median [IQR] and categorical variables as *n* (%). Differences between groups assessed by Kruskal–Wallis (continuous) or Fisher's exact test (categorical).

The SARS-CoV-2 Spike (S) protein is comprised of two subunits —S1 and S2, which are responsible for host cell receptor attachment and membrane fusion, respectively. There is growing evidence that antibodies against each subunit may contribute distinctly to immune protection[42,43], with anti-S1 IgG responses thought to have greater SARS-CoV-2 specificity and neutralizing capability, and anti-S2 IgG potentially reflecting memory B-cell immunity related to the greater conservation of S2 between coronaviruses[44–46]. To understand whether the different vaccine platforms elicited antibodies directed at different epitopes of Spike, we plotted the S1- and S2-specific IgG1 and FcR-binding in maternal plasma by the vaccine. Interestingly, the IgG1 titer directed against S1 was comparable across vaccine platforms, whereas women who received Ad26.COV2.S had a nonsignificant decrease in IgG1 titer against S2 (Fig. 1C). FcR-binding against the S1 domain was similar among vaccine platforms, whereas the FcR-binding against S2 was significantly lower for individuals who received Ad26.COV2.S (Fig. 1C). These data suggest that differences in the FcR-binding of antibodies against S2 are primary drivers of reduced Ad26.COV2.S functions against Spike. Moreover, these data highlight a single dose of Ad26.COV2.S can induce a similar S1-directed response as two doses of mRNA-1273 and BNT162b2.

To further examine differences in maternal vaccine response across vaccine platforms, a partial least squares discriminant analysis (PLSDA) was performed. The least absolute shrinkage

and selection operator (LASSO) was used to select features most important to the model to prevent overfitting. This analysis revealed that although the vaccine responses in individuals who received mRNA-1273 or BNT162b2 were indistinguishable, the vaccine response in individuals who received Ad26.COV2.S was clearly separated from those who received either mRNA vaccine (Fig. 1D). We next analyzed the enrichment of the LASSO-selected features (Fig. 1E) in each group, with nearly all LASSO-selected features higher in the women who received mRNA vaccination (Fig. 1E).

Finally, given the role of vaccine-induced neutralizing antibodies in providing protection from infection against SARS-CoV-2[47,48], we assessed the ability of the different vaccine platforms to induce neutralizing antibodies. Our group and others have previously demonstrated the induction of high titers of neutralizing antibodies to ancestral Spike after mRNA vaccination in pregnancy[15,17,25]. Given that the Omicron variant has displaced all other variants of SARS-CoV-2[49], and therefore knowledge of neutralization against Omicron is most relevant at this time in the pandemic, we measured neutralization activity against an Omicron pseudovirus. We found those mRNA vaccines induced neutralizing antibodies in a greater proportion of pregnant individuals than did the Ad26.COV2.S vaccine, although this difference did not reach statistical significance (52% (13/25) for mRNA-1273, 35% (12/34) for BNT162b2, 18% (2/11)

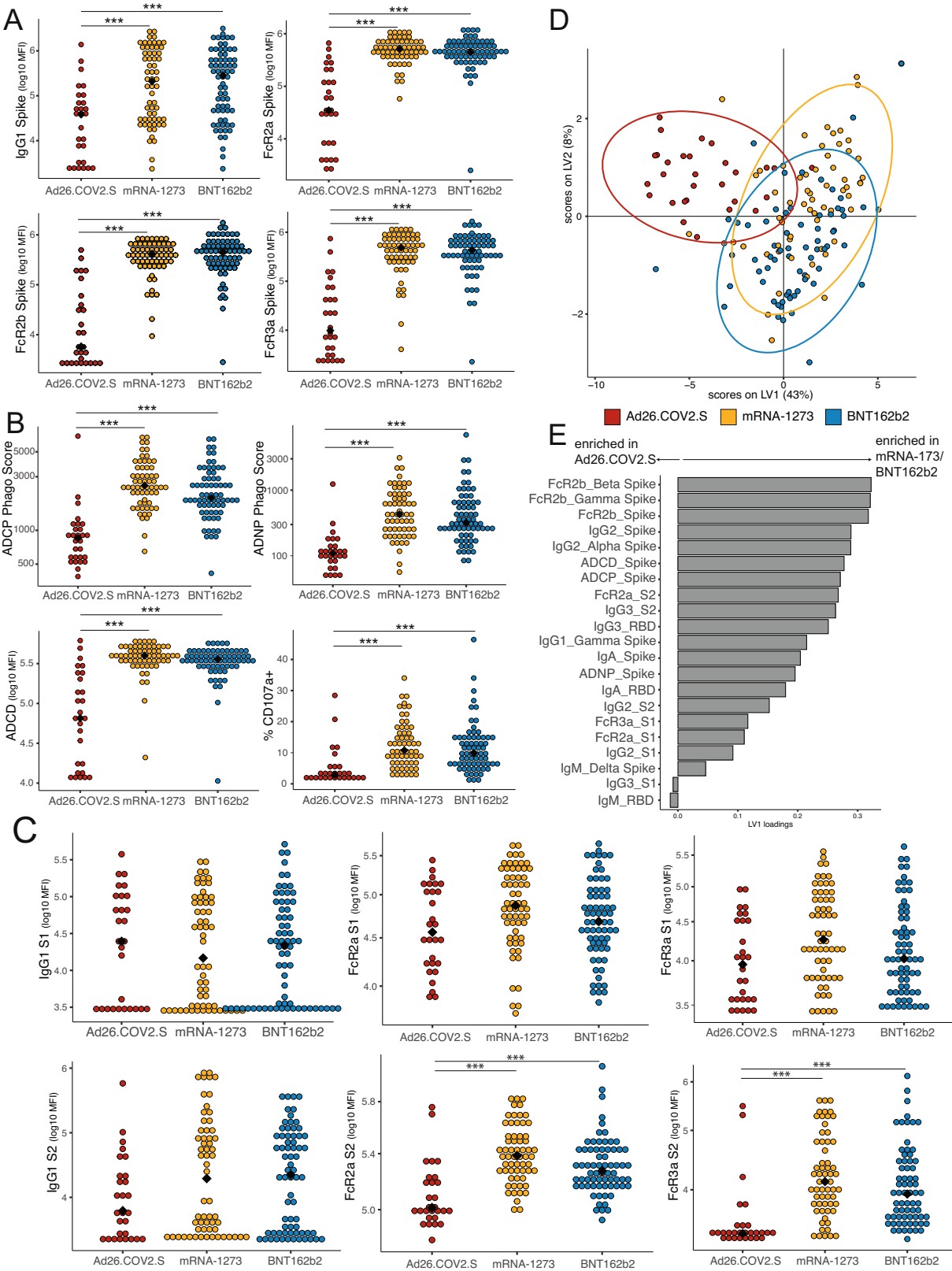

for Ad26.COV2.S, $p = 0.13$, Fig. S2B). The trimester of vaccination did not have a significant impact on the proportion of individuals with neutralizing antibodies against Omicron pseudovirus, with 50% (7/14) in the first trimester, 29% (8/28) in the second trimester, and 43% (12/28) of third trimester samples demonstrating neutralizing activity against Omicron ($p = 0.34$,

Fig. S2C). While the proportion of individuals with neutralizing activity against Omicron (39%) is lower than what has been previously reported against the ancestral virus[15,17,50], the reduced neutralizing activity against Omicron observed here is consistent with studies in non-pregnant individuals and limited available data in pregnancy[51–55].

**Fig. 1 Maternal vaccine-induced titers are comparable between mRNA-1273 and BNT162b2 vaccination, but lower after Ad26.COV2.S vaccination.**
**A** Spike-specific IgG1 and Fc-receptor (FcR) binding were measured by Luminex. The dot plots show the titer for pregnant individuals who received Ad26.COV2.S (red), mRNA-1273 (yellow) or BNT162b2 (blue). The black diamond represents the group median. Significance was determined by Kruskal–Wallis test followed by posthoc Benjamini–Hochberg correction adjustment, *$p < 0.05$, **$p < 0.01$, ***$p < 0.001$. The exact $p$-values stated in the following order for all subpanels: mRNA-1273 vs Ad26.COV2.S, then BNT162b2 vs Ad26COV2.S. IgG1: $p = 0.00035$; $p = 0.00048$. FcR2a: $p = 0.00018$; $p = 0.00019$. FcR2b: $p = 0.00018$; $p = 0.00019$. FcR3a: $p = 0.00018$; $p = 0.00019$. **B** The dot plots show the Spike-specific antibody-dependent cellular phagocytosis (ADCP), antibody-dependent neutrophil phagocytosis (ADNP), antibody-dependent complement deposition (ADCD), and antibody-dependent NK cell degranulation, as measured by % CD107a + NK cells, in maternal samples. The black diamond represents the group median. Significance was determined by Kruskal–Wallis test followed by posthoc Benjamini–Hochberg correction adjustment, *$p < 0.05$, **$p < 0.01$, ***$p < 0.001$. The exact $p$-values stated in the following order for all subpanels: mRNA-1273 vs Ad26.COV2.S, then BNT162b2 vs Ad26COV2.S. ADCP: $p = 0.00018$ for both; ADNP: $p = 0.0001$ both; ADCD: $p = 0.0001$ both; % CD107a+: $p = 0.00019$ for both. **C** The dot plots show the S1- (top) or S2- (bottom) specific IgG1 or FcR-binding in maternal samples. Significance was determined by Kruskal–Wallis test followed by posthoc Benjamini–Hochberg correction adjustment, *$p < 0.05$, **$p < 0.01$, ***$p < 0.001$. The black diamond represents the group median. The exact $p$-values stated in the following order for all subpanels: mRNA-1273 vs Ad26.COV2.S, then BNT162b2 vs Ad26COV2.S. FcR2a S2: $p = 0.00018$, $p = 0.0006$. FcR3a S2: $p = 0.000045$, $p = 0.000036$. **D** A partial-least squares discriminant model (PLSDA) was built using least absolute shrinkage and selection operator (LASSO)-selected SARS-CoV-2 specific antibody features in maternal samples, using vaccine type as the outcome variable. Each dot represents a sample, with the color representing the vaccine type. The ellipses represent the 95% confidence interval for the vaccine. **E** The barplot shows the latent variable (LV) 1 for the LASSO-selected features for the PLSDA in (**D**). Features with a positive loading along LV1 are enriched in mothers who received an mRNA vaccination, and features with a negative loading are enriched in mothers who received Ad26.COV2.S. Nearly all features are enriched in mRNA vaccine recipients. Source data are presented in Source Data File 1.

**Maternal immune response by trimester of vaccination.** We next sought to determine how the trimester of vaccination impacts the maternal vaccine-induced antibody response. Univariate analyses examining responses by trimester did not reveal trimester-specific differences in anti-Spike antibodies or FcR-binding (Fig. 2A, B, Fig. S3A, B). To further investigate the relative contribution of the trimester of vaccination to anti-Spike antibody titer, FcR-binding, and function, the mean percentile rank of each feature was plotted by trimester of vaccination (Fig. 2C). This analysis revealed that both first and third trimester vaccination drove a higher, albeit not statistically significant, functional antibody response compared to second trimester vaccination, marked by both higher FcR-binding and more functional antibodies as indicated by enhanced ADCD, ADNP, ADCP, and ADNKA responses.

Given the observed differences in immune response driven by vaccine platform, we next sought to define the combination of features that best separate vaccine responses by trimester of vaccination within each vaccine platform group. To this end, LASSO was used to pick a minimal set of features that differentiated individuals vaccinated in the second and third trimesters, followed by PLSDA to visualize the separation between the second and third trimesters (Fig. 2D); due to the smaller sample size, first trimester responses were not included in these analyses. Whereas there was little separation between second and third trimester vaccine responses in women that received the Ad26.COV2.S vaccine (5-fold CV: 0.3), there was a clear separation between the trimesters in women that received the mRNA-1273 vaccines (5-fold CV: 0.89, $p < 0.05$) and a modest separation between the trimesters in women that received the BNT162b2 vaccine (5-fold CV: 0.73, $p < 0.05$). The LASSO-selected features show enrichment of antibody measurements in the third trimester relative to the second within mRNA vaccine groups (Fig. S3C, D). Specifically, individuals who received mRNA-1273 during the third trimester had an enrichment in IgA and IgG2 against variants of concern Alpha and Beta, and enrichment of ADCP compared to those who received mRNA-1273 in the second trimester (Fig. S3C). This elevation in the IgA and IgG2 response in mRNA-1273 recipients was linked to a highly correlated response across SARS-CoV-2 variants (Fig. S3C), and the increase in ADCP was strongly correlated with FcR-binding across variants of concern and ADNP activity in these women (S3C). Women who received BNT162b2 in the

third trimester had enriched FcR2b-binding and IgM against the Alpha variant, and enriched IgG3 and ADNKA (measured by CD107a expression) responses compared to women who received BNT162b2 in the second trimester (Fig. S3D). The increase in Alpha FcR2b-binding seen in third trimester BNT162b2 recipients were highly correlated with FcR-binding and IgG3 titer across SARS-CoV-2 variants, showing that these antibodies are highly inflammatory and likely highly functional.

**Transplacental antibody transfer by vaccine platform.** To assess differences in the vaccine-induced immune response transferred from maternal to fetal circulation by vaccine type, we plotted the Spike-specific antibody titer and FcR-binding in umbilical cord serum in the 123 dyads who underwent systems serology profiling. In the cord blood, Spike-specific antibody titers and Fc-receptor binding were significantly higher in recipients of mRNA-1273 or BNT162b2 compared to recipients of Ad26.COV2.S (Fig. 3A and Fig. S4A). IgG2 against Spike was significantly higher in the cord blood of mRNA-1273 recipients compared to either Ad26.COV2.S or BNT162b2 recipients (Fig. S4A). Moreover, Spike-specific antibody titers and functions in cord blood were highly correlated with the antibody response against variants of concern across all vaccine platforms, suggesting that Spike-specific antibodies in cord blood are likely to be active against variants of concern (Fig. S4B). These observed differences in Spike-specific antibody response and response to variants of concern by vaccine platform in umbilical cord blood mirrored those observed in the maternal antibody response.

Functional antibody responses, including ADCP, ADNP, ADCD, and ADNKA (measured as % CD107a+, % MIP-1β+ and % IFNγ+ cells), were lower in the cord blood of Ad26.COV2.S recipients compared to mRNA-1273 or BNT162b2 (Fig. 3B and Fig. S4C). Interestingly, the ADCP and ADCD responses in the cord blood of women who received mRNA-1273 were significantly higher than those of women who received either Ad26.COV2.S or BNT162b2 (Fig. 3B), whereas the response in the maternal blood was similar between those two vaccines (Fig. 1B and Fig. S1B), suggesting preferential transfer of these highly functional antibodies in mRNA-1273 recipients. Similar to what was observed in the maternal blood, vaccination with Ad26.COV2.S resulted in equivalent IgG1 titer and FcR-binding against S1 in cord blood compared to mRNA

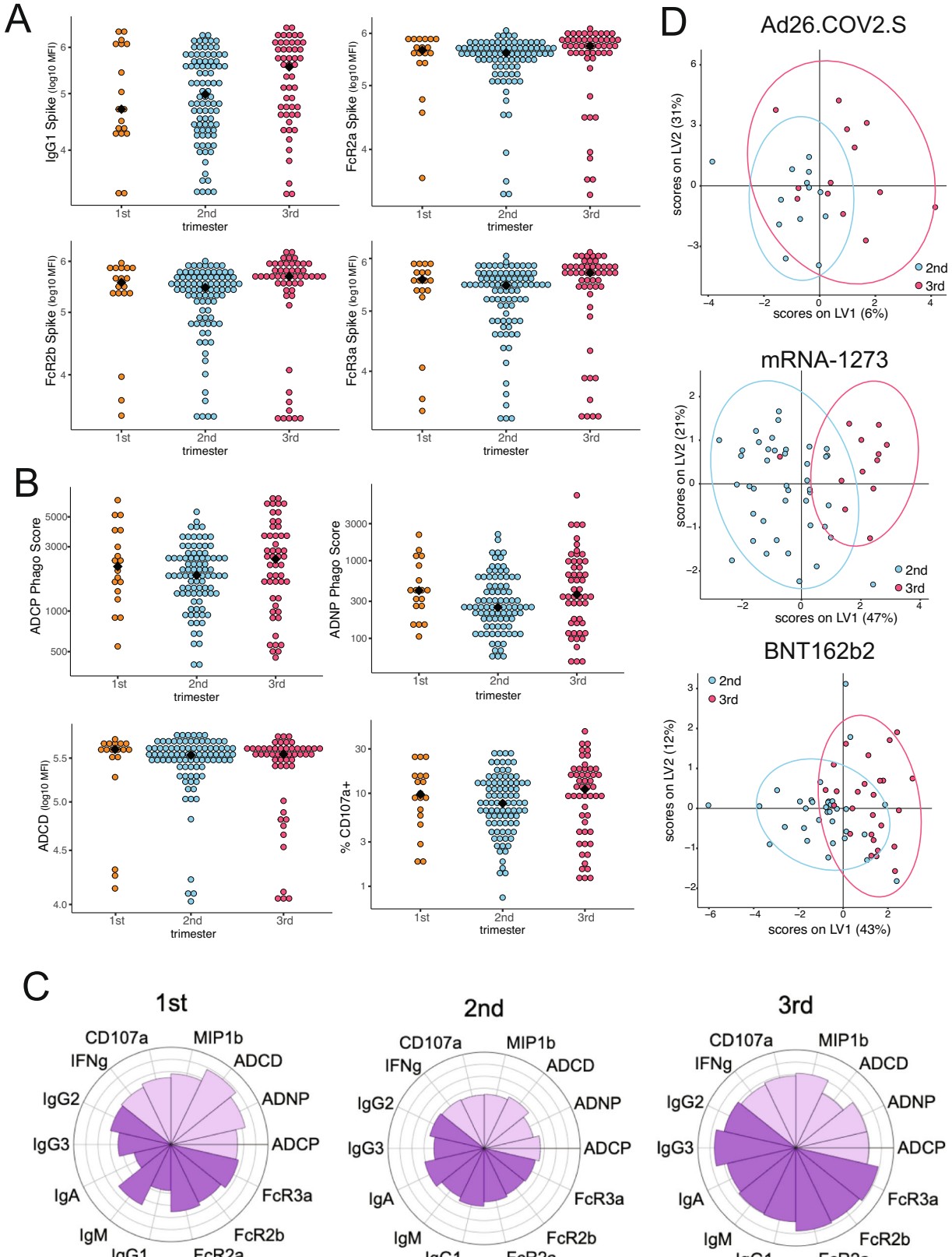

vaccination, but significantly lower FcR-binding antibodies (FcR2a, FcR2b, and FcR3a) against S2 in cord blood (Fig. 3C).

A LASSO-PLSDA model was built using antibody features in cord blood to elucidate which antibody classes are enriched in the cord blood across vaccine platforms (Fig. 3D). Whereas the mRNA-1273 and BNT162b2 cord blood responses had significant

overlap, the Ad26.COV2.S cord blood response separated from the two other vaccine responses (Fig. 3D). Moreover, all LASSO-selected features were enriched in the cord blood of women who had received mRNA-1273 or BNT162b2 compared to Ad26.COV2.S (Fig. 3E). These data demonstrate strong similarities between maternal and cord blood antibody titers and

**Fig. 2 Trimester of vaccination affects vaccine-induced antibody titer in maternal samples. A** The dot plots show the Spike-directed IgG1 and FcR-binding in maternal samples by trimester of vaccination. Significance was determined by Kruskal–Wallis test. No significant differences were found. The black diamond represents the group median. **B** The dot plots show the Spike-directed antibody-dependent cellular phagocytosis (ADCP), antibody-dependent neutrophil phagocytosis (ADNP), antibody-dependent complement deposition (ADCD) and antibody-dependent NK cell degranulation, as measured by % CD107a + NK cells, in maternal samples by trimester of vaccination. Significance was determined by Kruskal–Wallis test. No significant differences were found. The black diamond represents the group median. **C** The polar plots show the mean percentile rank for Spike-specific features in the first, second, and third trimesters of vaccination. **D** A PLSDA was built using LASSO-selected antibody features in maternal plasma for mothers who received Ad26.COV2.S, mRNA-1273, or BNT162b2 using the trimester of vaccination as the outcome variable. Each dot represents a sample, with the color representing the trimester. The ellipses represent the 95% confidence interval for the trimester. Source data are presented in Source Data File 1.

functions, and reduced titer, FcR-binding and functionality of cord blood antibodies in recipients of Ad26.COV2.S relative to the mRNA vaccines.

To further probe the contribution of vaccine type to transplacental antibody transfer, we plotted the matched maternal-cord antibody titers and functions for each vaccine and compared differences (Fig. 4A, B). While we expect titers to be higher in the cord relative to maternal serum for most vaccine-induced antibodies[56–58], it was notable that no antibody feature was significantly higher in the cord blood of women who had received the Ad26.COV2.S vaccine (Fig. 4A, B). In contrast, nearly all Spike-specific antibody functions were higher in cord blood of women who received mRNA-1273 or BNT162b2 compared to maternal blood (ADCP, ADNP, and ADNKA by %CD107a+), with the exception of ADCD (Fig. 4B).

Given that transplacental transfer of antibodies is driven substantially by maternal titers[59,60], it is possible that the lower transfer of antibodies in women that received Ad26.COV2.S could simply be due to lower maternal titers after Ad26.COV2.S compared to the mRNA vaccines. To reveal whether different vaccine platforms result in an enrichment of different antibody features in the cord blood, we performed a multilevel PLSDA (mPLSDA) using LASSO to select features that were most different between maternal and cord blood for each vaccine (Fig. 4C–E, Fig. S4). This approach accounts for the heterogeneous responses between vaccine recipients at the individual level. All three vaccines showed separation between maternal and cord blood and enrichment of FcR-binding and Spike-specific IgG titer in the cord blood relative to maternal (Fig. 4C–E, Fig. S5). Interestingly, while we did not observe any significant differences between maternal and cord blood in Ad26.COV2.S vaccine recipients through univariate analysis, on a multivariate level, FcR2-binding antibodies, anti-S1 IgG2, and NK-cell-activating antibodies (MIP-1β) were enriched in cord blood of the Ad26.COV.S dyads, while maternal blood was enriched for anti-S2 IgG1 and IgG3, and ADCD (Fig. 4C, Fig. S5A). Thus, despite lower efficiency of antibody transfer in women that received Ad26.COV2.S, all three vaccines allowed for the preferential transfer of specific antibodies to the cord blood. Similarly, the cord blood of mRNA-1273 or BNT162b2 recipients was enriched for IgG1 and IgG2, FcR-binding antibodies, functional antibodies, and antibodies directed against variants of concern (Alpha, Beta, and Delta), whereas maternal blood was enriched for IgG3 (Fig. 4D, E, Fig. S5B, C). These data highlight that although slight differences in transfer efficiency exist between vaccines, placental enrichment for highly functional antibodies in the umbilical cord is a commonality that likely reflects a fundamental principle of transplacental transfer biology, consistent with prior work demonstrating that highly functional, NK cell-activating, FcR-binding antibodies are preferentially transferred from maternal to cord blood in response to vaccination[16,61].

**Transplacental antibody transfer by trimester of vaccination.** To investigate the impact of the trimester of vaccination on the

transplacental transfer of vaccine-induced immunity to the neonate at delivery, we measured total anti-Spike antibody IgG (as assessed by ELISA, see Methods) in the 123 dyads included in systems serology analyses and an additional 52 dyads who had delivered by study completion ($N = 175$ total dyads, Supplemental Table 2). Interestingly, anti-Spike antibody titers in umbilical cord blood were higher than maternal titers at delivery when vaccination occurred in the first and second but not third trimesters (Fig. 5A).

The transfer ratio (TR)—defined as the ratio of cord blood anti-Spike IgG titer to maternal anti-Spike IgG titer at delivery—was calculated for each maternal-neonatal dyad and plotted by trimester of vaccination as a metric of transfer efficiency (Fig. 5B). This analysis revealed higher TRs generated by first and second trimester vaccination (median TR = 1.5 and 1.3) compared to third trimester vaccination (median TR = 1.0). For reference, the expected efficiency of transplacental antibody transfer is >1, indicating higher cord titers at delivery compared to maternal titers, with TRs of 1.2–3 at delivery noted for other vaccine-induced titers, such as measles, influenza, and pertussis[56–58].

We next sought to determine absolute anti-Spike IgG titer in the cord blood at delivery by trimester of vaccination. Total anti-Spike IgG after first trimester vaccination was significantly lower than that in the cord blood of second trimester vaccine recipients (Fig. 5C). In the context of the highest TRs observed in first trimester vaccine recipients, this finding likely reflects a waning of maternal titers at delivery compared to second trimester vaccine recipients. Given the finding of highest TRs for first trimester vaccination but lower absolute titers in cords of mothers vaccinated in the first trimester, suggestive of high placental efficiency in the setting of maternal first trimester vaccination but waning maternal titers by delivery, we next sought to quantify the waning of maternal antibody titers in first and second trimester vaccine recipients from completion of vaccination to delivery. In the subset of dyads in whom blood was drawn at 2–6 weeks following the second vaccine dose in mRNA vaccine recipients or following the single Ad26.COV2.S dose, and again at delivery ($n = 7$ first trimester, $n = 19$ second trimester), we compared total maternal anti-Spike IgG post-boost (second dose) and at delivery. This analysis demonstrated that antibody titers were significantly lower at delivery compared to shortly after the boost dose in both first and second trimester vaccine recipients (Fig. 5D). As expected, the ratio of titers at delivery to post-boost was lower for first trimester vaccine recipients compared to second trimester vaccine recipients (Fig. 5E), likely reflecting a more significant waning of antibody titer with time since vaccination.

## Discussion

Pregnancy is a unique immunological epoch, requiring complex and trimester-specific alterations in the maternal immune response to both protect the maternal-neonatal dyad and promote maternal tolerance of the semi-foreign fetal allograft[32–34]. The COVID-19 pandemic revealed key deficits in our knowledge

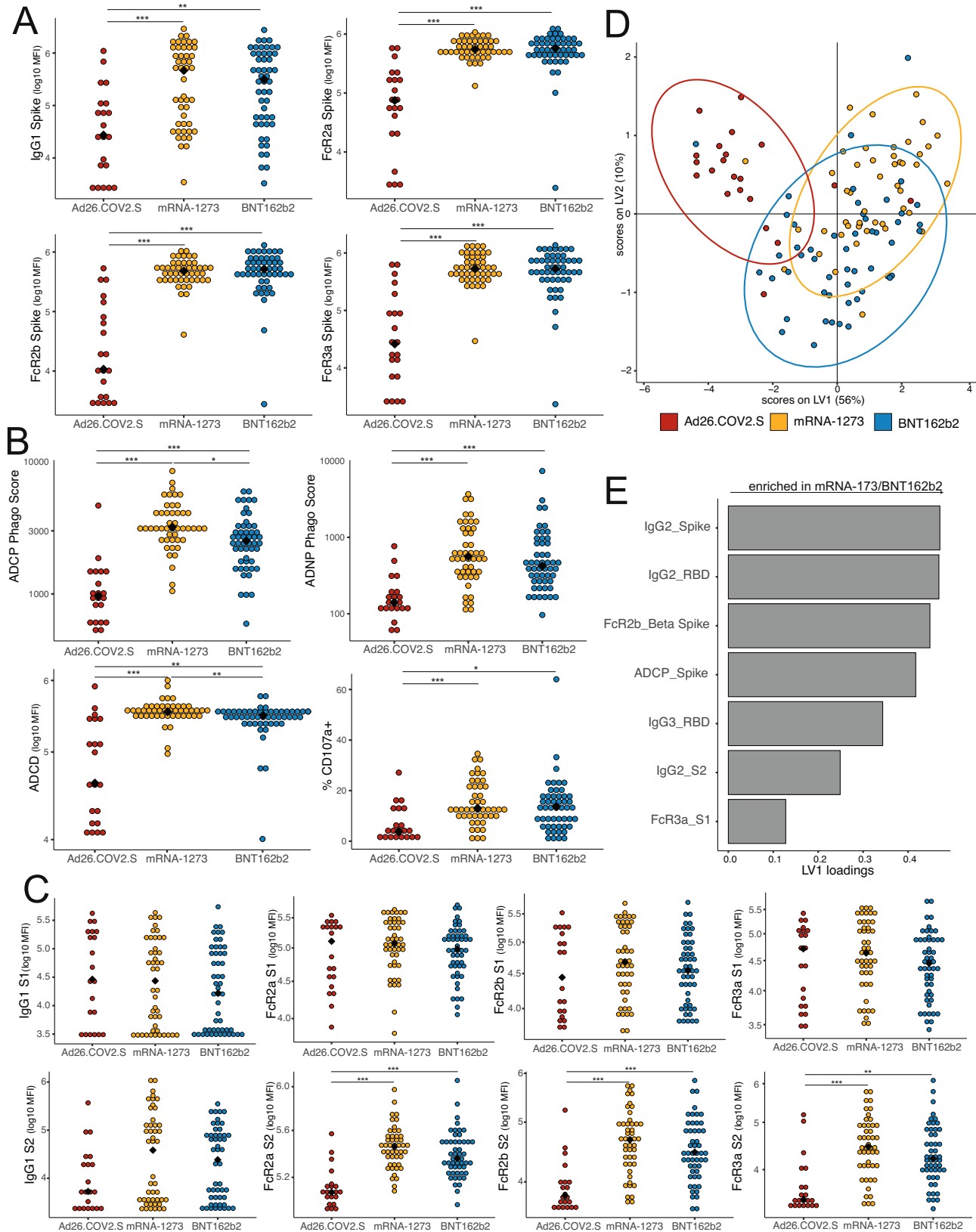

of both normal and challenged pregnancy immunity as well as the maternal response to vaccines. Although many vaccines can be safely administered in pregnancy, with seasonal influenza and tetanus-diphtheria-acellular pertussis (Tdap) routinely recommended for all pregnant individuals[31,62], observational studies of vaccination in pregnancy have generated only a limited understanding of how vaccine and pregnancy characteristics interact to impact vaccine-induced immune profiles[30,63–66]. Here, we present evidence of both platform- and trimester-specific differences in the maternal immune response to the three COVID-19 vaccines available in the United States. We demonstrate that while functional Spike-specific maternal antibodies are

**Fig. 3 Vaccine-induced antibody titer in cord samples is comparable between mRNA-1273 and BNT162b2 vaccination but lower after Ad26.COV2.S vaccination. A** Spike-specific IgG1 and Fc-receptor (FcR) binding were measured by Luminex. The dot plots show the titer for cords whose mothers who received Ad26.COV2.S (red), mRNA-1273 (yellow) or BNT162b2 (blue). Significance was determined by Kruskal–Wallis test followed by posthoc Benjamini–Hochberg correction adjustment, *$p < 0.05$, **$p < 0.01$, ***$p < 0.001$. The black diamond represents the group median. Exact p-values stated in the following order for all subpanels: mRNA-1273 vs Ad26.COV2.S, then BNT162b2 vs Ad26COV2.S. IgG1: $p = 0.001$, $p = 0.0031$. FcR2a: $p = 0.00036$, $p = 0.00018$. FcR2b: $p = 0.00012$, $p = 0.00009$. FcR3a: $p = 0.000072$, $p = 0.00006$. **B** The dot plots show the Spike-specific antibody-dependent cellular phagocytosis (ADCP), antibody-dependent neutrophil phagocytosis (ADNP), antibody-dependent complement deposition (ADCD) and antibody-dependent NK cell degranulation, as measured by % CD107a + NK cells, in cord samples. Significance was determined by Kruskal–Wallis test followed by posthoc Benjamini–Hochberg correction adjustment, *$p < 0.05$, **$p < 0.01$, ***$p < 0.001$. The black diamond represents the group median. Exact p-values stated in the following order for all subpanels: mRNA-1273 vs Ad26.COV2.S, then BNT162b2 vs Ad26COV2.S, then mRNA-1273 vs BNT162b2 (if relevant). ADCP: $p = 0.000036$, $p = 0.000033$, $p = 0.02$. ADNP: $p = 0.00003$, $p = 0.000028$. ADCD: $p = 0.000026$, $p = 0.005$, $p = 0.0025$. CD107a+: $p = 0.00085$, $p = 0.006$. **C** The dot plots show the S1 (top) or S2 (bottom) specific IgG1 or FcR-binding in cord samples. Significance was determined by Kruskal–Wallis test followed by posthoc Benjamini–Hochberg correction adjustment, *$p < 0.05$, **$p < 0.01$, ***$p < 0.001$. The black diamond represents the group median. Exact p-values stated in the following order for all subpanels: mRNA-1273 vs Ad26.COV2.S, then BNT162b2 vs Ad26COV2.S. FcR2a S2: $p = 0.00045$ for both. FcR2b S2: $p = 0.00045$, $p = 0.00072$. FcR3a S2: $p = 0.00045$, $p = 0.0027$. **D** A partial-least squares discriminant model (PLSDA) was built using LASSO-selected SARS-CoV-2 specific antibody features in cord samples, using vaccine type as the outcome variable. Each dot represents a sample, with the color representing the vaccine type. The ellipses represent the 95% confidence interval for the vaccine. **E** The barplot shows the latent variable (LV) 1 for the least absolute shrinkage and selection operator (LASSO)-selected features for the PLSDA in (**D**). Features that with a positive LV1 loading were enriched in the cords whose mothers received an mRNA vaccine. Source data are presented in Source Data File 1.

generated by all vaccine platforms, antibody titers and functionality profiles are enhanced in response to mRNA vaccines when compared with Ad26.COV2.S, with mRNA-1273 demonstrating subtle functional advantages over BNT162b2. Vaccination in the first and third trimesters induced greater immunogenicity when compared with second trimester vaccination, with further analysis of mRNA vaccine recipients indicating less functionality against variants of concern in the second trimester versus third trimester vaccination. Although total Spike-specific antibody titers were lower in the umbilical cord at delivery following first trimester vaccination, possibly due to waning maternal antibody titers over time, we observed the highest transfer efficiency of functional antibodies from mother to umbilical cord following first trimester vaccination.

Assessment of maternal and cord humoral immunity across vaccine platforms demonstrated significantly higher Spike-specific antibody titers, FcR-binding, and functional antibodies induced by both mRNA COVID-19 vaccines compared to those induced by the Ad26.COV2.S vaccine. Although Ad26.COV2.S vaccination induces a lower antibody response than the mRNA vaccines in pregnant individuals, as has been observed in non-pregnant individuals[39,67–69], the majority of participants who received Ad26.COV2.S in pregnancy still had detectable antibody titers and functions after a single vaccine dose. Recent studies have underscored the importance of central and effector memory T-cell responses in Ad26.COV2.S recipients[70,71] and improved antibody coverage and neutralizing capacity against variants of concern over the eight months following single vaccination[72], suggesting that maturation of B cell responses occurs in Ad26.COV2.S recipients without boosting and is an important driver of antibody protection over time. Thus, our assessment of antibody titers and functions in these Ad26.COV2.S recipients at a median of 44 days post vaccine may not capture the full depth and breadth of the immune response. The observed differences in maternal antibody quantity and quality between Ad26.COV2.S vaccine recipients and mRNA vaccine recipients might be a reflection of a one- versus two-dose regimen, rather than a reflection of inferior response to the Ad-vectored vaccine platform itself. This concept is supported by recent data suggesting that the Ad26.COV2.S vaccine protection is enhanced in a two-dose regimen[73,74], with "booster" dose recommended any time two months or more after initial dose to enhance protection in specific vulnerable populations[75]. Further evaluation of whether the differences seen between mRNA vaccines and the Ad26.COV2.S vaccine persists after two doses of the Ad-vectored

vaccine is a critical area for future research, and will elucidate whether the differences noted in pregnancy relate simply to dosing and interval, versus the Ad-vectored platform itself. As previous work has shown the importance of adherence to the prime/boost timeline for mRNA vaccine recipients given delayed kinetics of antibody responses during pregnancy[16], investigating the impact of "booster" doses given during pregnancy, particularly in recipients who originally received the Ad26.COV2.S vaccine, will be important to obtain a full understanding of how vaccine strategies impact the maternal immune response and antibody transfer to the neonate.

Robust changes in the inflammatory profile occur during pregnancy to facilitate implantation and early placentation, followed by a period of rapid fetal growth, and finally, the onset of parturition[32–35]. How these immune fluctuations influence maternal responses to vaccines administered across gestation is not known. Although univariate analyses examining responses by trimester did not reveal trimester-specific differences, these analyses likely fail to account for interactions between multiple elements of the antibody response. Harnessing the strength of the systems serology approach, we identified increased immunogenicity— characterized by anti-Spike antibody FcR-binding capacity and functionality—in pregnant individuals vaccinated in the first and third trimesters compared to those vaccinated in the second trimester through multivariate modeling. These data demonstrate the importance of considering more than just Ig titer and neutralization alone when evaluating vaccine immunogenicity, particularly with the rise of variants of concern that demonstrate significant escape from vaccine-induced neutralizing antibodies[51–54]. Further investigation into the anti-Spike antibody responses (including against variants of concern) in second and third trimester vaccine recipients by vaccine type revealed that second trimester responses are impaired compared with the third trimester for both mRNA vaccines. Taken together, these data suggest that second trimester vaccination generates an antibody response characterized by overall reduced FcR-binding capacity and functionality relative to vaccination in the first and third trimesters. These findings can be understood in the context of immunomodulatory changes that occur in the second trimester of pregnancy that favor maternal tolerance of the developing fetal semi-allograft and promote a state of immunological quiescence[35], in which response to non-self antigens may be dampened.

While effective maternal protection is paramount during a pandemic, neonatal protection against potentially harmful

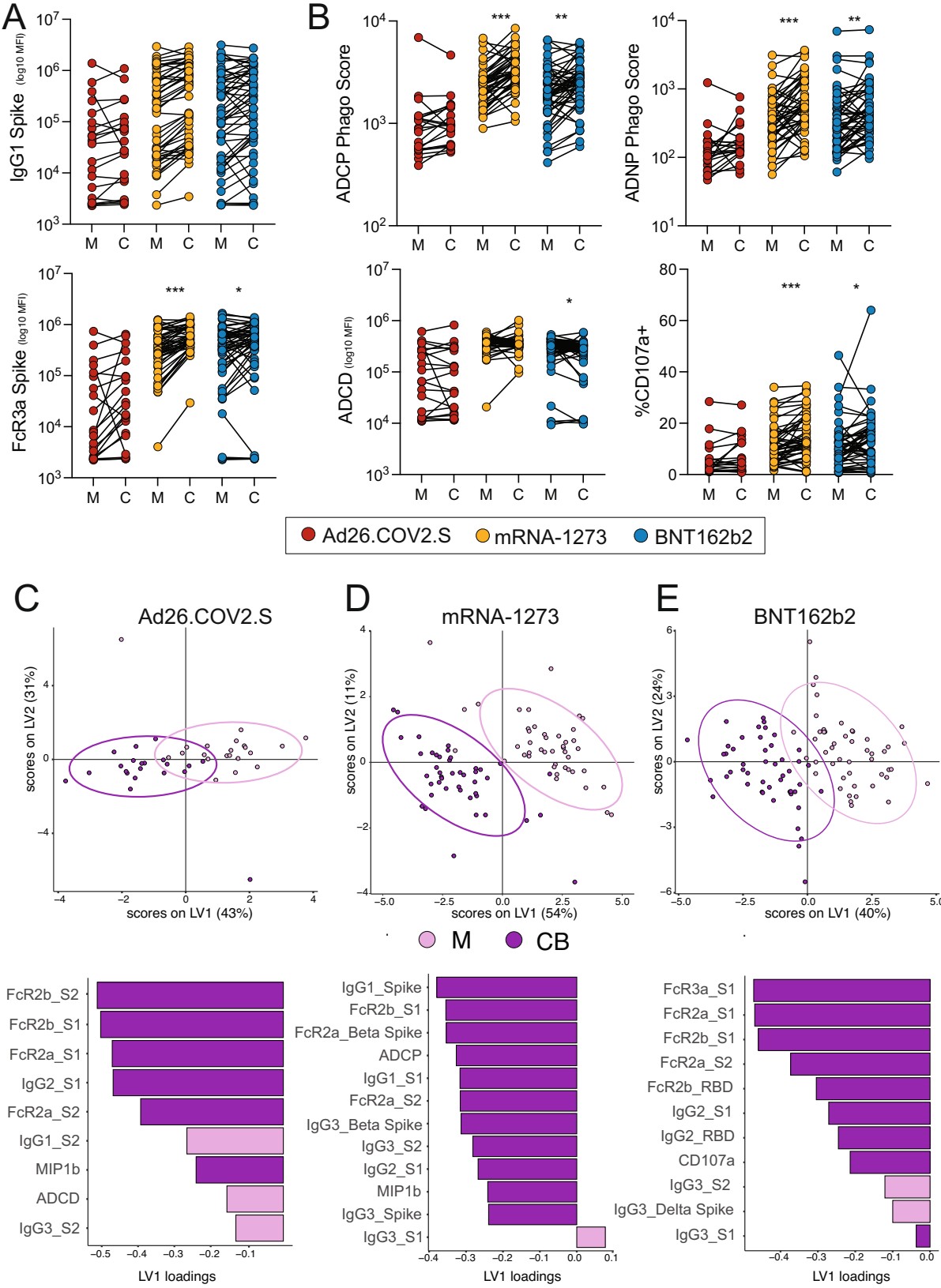

pathogens through maternal immunization is an important sec-ondary consideration when developing vaccine recommendations in pregnancy[30,31]. Recipients of all 3 COVID-19 vaccines demonstrated enrichment of FcR-binding and anti-Spike IgG titer in the cord compared with maternal blood, with the most favorable cord:maternal transfer ratios (>1) following first and

second trimester vaccination. Whether higher antibody transfer ratios observed for first trimester COVID-19 vaccination are due to differences in antibody Fc-quality and thus affinity for Fc-receptors that traffic antibody to the fetal circulation[60,61,76], or are the result of increased time for antibody transit to occur, or a combination of both is yet to be determined. As expected, the

**Fig. 4 Efficient transfer of vaccine-induced antibodies to cord blood. A** The dot plots show the Spike-specific IgG1 titer or FcR3a binding for maternal plasma (M) and cord blood (C). Lines connect maternal-cord dyads and the color represents vaccine type, Ad26.COV2.S (red), mRNA-1273 (yellow) or BNT162b2 (blue). Significance was determined by Wilcoxon signed-rank test (2-sided) followed by posthoc Benjamini–Hochberg correction adjustment, *$p < 0.05$, **$p < 0.01$, ***$p < 0.001$. Exact $p$-value for FcR3a: mRNA-1273 $p = 0.0000052$, BNT162b2 $p = 0.042$. **B** The dot plots show the Spike-specific antibody-dependent cellular phagocytosis (ADCP), antibody-dependent neutrophil phagocytosis (ADNP), antibody-dependent complement deposition (ADCD), and antibody-dependent NK cell degranulation, as measured by % CD107a + NK cells, for maternal plasma (M) and cord blood (C). Lines connect maternal-cord dyads and the color represents vaccine type, Ad26.COV2.S (red), mRNA-1273 (yellow) or BNT162b2 (blue). Significance was determined by Wilcoxon signed-rank test (2-sided) followed by posthoc Benjamini–Hochberg correction adjustment, *$p < 0.05$, **$p < 0.01$, ***$p < 0.001$. Exact $p$-values stated in the following order for all subpanels: mRNA-1273 M-C, then BNT162b2 M-C, unless otherwise noted. ADCP: $p = 0.00011$, $p = 0.0066$. ADNP: $p = 0.00007$, $p = 0.0099$. ADCD: BNT162b2 $p = 0.011$. CD107a+: $p = 0.00021$, $p = 0.047$. **C–E** A multilevel PLSDA (mPLSDA) was built for Ad26.COV2.S (**C**), mRNA-1273 (**D**) and BNT162b2 (**E**) using sample type, maternal blood (M, light purple) or cord blood (CB, dark purple) as the outcome variable. Features were selected using LASSO prior to building the models. The dot plots (top) show the scores plots for the mPLSDA. Each dot represents a sample, with the color representing the sample type. The ellipses represent the 95% confidence interval for the sample type. The bar plots show the LV1 for the mPLSDA built in each respective subfigure. Source data are presented in Source Data File 1.

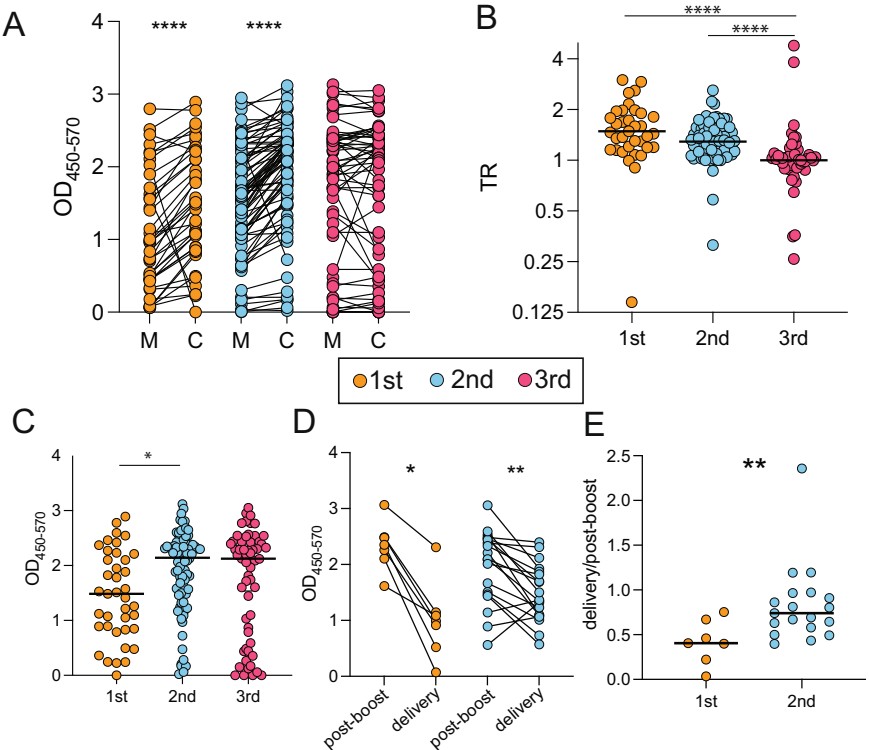

**Fig. 5 Transfer efficiency differs by trimester of vaccination. A** The dot plots show the Spike-specific IgG titer in maternal (M) or cord (C) plasma. Lines connect paired dyads. The color indicates the trimester of vaccination, first (orange), second (blue) or third (pink). Significance was determined by a Wilcoxon signed-rank test (2-sided), *$p < 0.05$, **$p < 0.01$, ***$p < 0.001$, ****$p < 0.0001$. Exact $p$-values presented first trimester: $p < 0.0001$ (GraphPad Prism does not provide a more exact $p$-value for this comparison), second trimester: $p < 0.0001$. **B** The dot plot shows the transfer ratio (cord titer/ maternal titer) of Spike-specific IgG to the cord. Color indicates the trimester of vaccination. Significance was determined by Kruskal–Wallis test. ****$p < 0.0001$. The horizontal line represents the group median. Exact $p$-value 1st vs 3rd TR: $p < 0.0001$ (GraphPad does not provide any more exact $p$-value); 2nd vs 3rd $p < 0.0001$. **C** The dot plot shows the Spike-specific IgG titer in cord blood by trimester that the mother received COVID-19 vaccination. Significance was determined by a one-way ANOVA followed by correction for multiple comparisons, *$p < 0.05$. The horizontal line represents the group median. Exact $p$-value 1st vs 2nd $p = 0.017$. **D** The dot plots show the IgG Spike titer in maternal plasma post-boost (~2–6 weeks after the second dose of mRNA vaccine or after a single dose of Ad26.COV2.S vaccine) and at delivery following vaccination in the first (orange) and second (blue) trimester. Lines connect matched samples. Significance was determined by a Wilcoxon signed-rank test (2-sided), *$p < 0.05$, **$p < 0.01$. Exact $p$-value 1st $p = 0.016$. Exact $p$-value 2nd $p = 0.002$. **E** The dot plot shows the ratio of the IgG Spike titer delivery/post-boost following vaccination in the first (orange) and second (blue) trimesters. Significance was determined by a two-tailed Mann–Whitney test, **$p < 0.01$. The horizontal line represents the group median. Exact $p$-value 1st vs 2nd $p = 0.0085$. Source data are presented in Source Data File 2.

waning of maternal titers was more significant after first compared to second trimester maternal vaccination, and both maternal and neonatal immunity would likely be boosted by a third maternal COVID-19 vaccine dose in the third trimester when primary vaccination (or vaccination series) occurs in the first trimester.

These results augment our current understanding of how the timing of maternal vaccination impacts both maternal immune response and transplacental transfer efficiency. Current clinical recommendations governing the timing of routinely administered vaccines in pregnancy (e.g., influenza vaccine, which is administered during influenza season regardless of trimester, and

Tdap, which is administered in the late second to early third trimester with the primary goal of enhancing transplacental transfer) have limited the ability to systematically investigate the impact of vaccine administration across gestation. The Advisory Committee on Immunization Practices (ACIP) has advised routine Tdap administration during pregnancy after 20 weeks of gestation for over a decade, based on the limited availability of safety data in the first trimester[77]. Studies that have investigated the timing of Tdap vaccination after the first trimester has found the superior transfer of anti-pertussis antibody when vaccination occurs earlier in the recommended interval of 27–36 weeks[65,78,79], with some evidence from individuals vaccinated outside that window favoring improved transfer following early second trimester vaccination[80], and other evidence showing no effect of gestational age on antibody transfer[76]. Data on placental transfer of anti-pertussis antibody following Tdap administration in the first trimester are not available, as studies including first trimester vaccinees are primarily limited to safety reports[81,82]. Data from seasonal influenza vaccine administration in pregnancy, which is administered at any gestational age during influenza season, are conflicting with respect to maternal immune response. Some studies suggest lower maternal anti-influenza titers in first compared to second trimester vaccination, with highest anti-influenza titers in third trimester vaccination[66,83,84], while others suggest a more robust maternal titer generated by first and third trimester vaccination relative to second[85]. In addition, a majority of studies noted enhanced cord blood antibody titers against influenza following third trimester vaccination when compared with second or first trimester vaccination, likely due in part to waning maternal antibody titers with increased time from vaccination[66]. These studies were limited by their narrow focus on IgG titer as the primary measure of the maternal immune response, while our systems serology approach permits the dissection of diverse components of maternal humoral immunity.

While our study was not designed to correlate serological data with clinical vaccine effectiveness outcomes and therefore cannot demonstrate antibody-mediated protection against severe disease, data from large epidemiologic studies have demonstrated the effectiveness of COVID-19 vaccines in protecting pregnant people from severe/critical COVID-19 and maternal mortality[9,20–24]. In addition, available data demonstrate that maternal COVID-19 vaccination is effective in preventing COVID-19 hospitalization in infants up to 6 months of age[29], likely due to the persistence of maternal antibodies in the infant for up to 6 months[86]. Importantly, for unvaccinated individuals who become pregnant, the CDC, the American College of Obstetricians and Gynecologists (ACOG), and the Society for Maternal-Fetal Medicine (SMFM) recommend vaccination as soon as possible, including in the first trimester[75,87,88] to maximize the amount of time during which the mother and fetus are protected from the harms of COVID-19 infection during pregnancy, which includes severe and critical maternal illness[20–23] as well as poor neonatal outcomes resulting from spontaneous or medically indicated preterm birth or stillbirth[7–10]. The data presented here demonstrate a comparable, if not increased, first trimester maternal immune response, as well as enrichment of functional antibodies in the cord and high transfer efficiency following first trimester vaccination. For pregnant individuals vaccinated in the first trimester, both maternal and neonatal immunity may be further enhanced by boosting in the third trimester[89], with boosting 6 months post-mRNA vaccines and two months post-Ad26.COV2.S vaccine is now recommended by the CDC, ACOG, and SMFM[75,87,88].

The rapid development and distribution of three COVID-19 vaccines in the US have offered an unprecedented opportunity to further our understanding of the rules of vaccine-induced immunity in pregnancy. Our study contributes to understanding how the maternal-neonatal dyad responds to vaccination against a de novo antigen with novel mRNA and Ad-vectored vaccines, which were not specifically designed to optimize maternal or neonatal protection, as pregnant individuals were excluded from initial vaccine clinical trials[12–14]. Looking beyond responses to the COVID-19 vaccines, our findings may have broader implications. These insights into maternal-neonatal dyad antibody-mediated immunity generated by COVID-19 vaccines may be used to guide rational vaccine development and administration[90], as efforts to define serological correlates of protection against COVID-19 continue to expand[91]. Recruitment and inclusion of pregnant individuals in vaccine studies will remain critical to constructing evidence-based vaccine strategies that maximize the benefit to both mother and newborn.

## Methods

**Participant recruitment and study design.** Pregnant individuals at two tertiary care centers were approached for enrollment in the COVID-19 pregnancy biorepository study between January 2021 and September 2021, Protocol #2020P003538, approved by Mass General Brigham Institutional Review Board (IRB). Eligible participants were pregnant, greater than or equal to 18 years old, able to provide informed consent, and received the Ad26.COV2.S, mRNA-1273, or BNT162b2 COVID-19 vaccine during pregnancy. Eligible participants were identified by practitioners at the participating hospitals or were self-referred. A study questionnaire was administered to assess pregnancy status, history of prior SARS-CoV-2 infection, the timing of COVID-19 vaccine doses, and type of COVID-19 vaccine received. Individuals were grouped by the type of vaccine received, and by the trimester at which the first vaccine dose was given in recipients of mRNA-1273 and BNT162b2 vaccines (or at the time of the single dose in recipients of the Ad26.COV2.S vaccine). To maximize the generalizability of our findings to the general pregnant population, participants who tested positive for SARS-CoV-2 prior to or after receiving a COVID-19 vaccine were not excluded from this study.

**Sample collection.** For Ad26.COV2.S vaccine recipients, blood was collected at least 2 weeks after receiving the single vaccine dose. For mRNA-1273 and BNT162b2 vaccine recipients, blood was collected at least 2 weeks following the second vaccine dose. For participants who delivered during the study time frame ($N = 123$), maternal blood was drawn at the time of delivery, and umbilical cord blood was collected after delivery. Blood was collected by venipuncture (or from the umbilical vein following delivery) into serum separator and EDTA tubes and centrifuged at $1000 \times g$ for 10 min at room temperature. Serum and plasma were aliquoted into cryogenic vials and stored at $-80\,°C$.

**Antigens.** Antigens used for assays included SARS-CoV-2 D614G Spike, Alpha Spike, Beta Spike, Gamma Spike, and Delta Spike (all Spikes kindly provided by Erica Ollman Saphire) and SARS-CoV-2 S1 and S2 (Sino Biological).

**Primary cells.** Neutrophils were isolated from fresh peripheral whole blood collected at the Ragon Institute. NK cells were isolated from fresh peripheral blood from buffy coats collected at Massachusetts General Hospital (MGH). All volunteers gave signed, informed consent and were over the age of 18, and samples were deidentified before use. The study was approved by the MGH Institutional Review Board. Neutrophils were maintained in R10 media (RPMI supplemented 10% fetal bovine serum (FBS) (Sigma Aldrich), 5% penicillin/streptomycin (Corning, 50 μg/mL), 5% L-glutamine (Corning, 4 mM), 5% HEPES buffer (pH 7.2) (Corning, 50 mM)) and at $37\,°C$, 5% $CO_2$ for the duration of the assay. After isolation, NK cells were rested overnight at R10 media supplemented with 2 ng/mL interleukin (IL)-15 at $37\,°C$, 5% $CO_2$.

**Bead-based functional assays.** For antibody-dependent cellular phagocytosis (ADCP), antibody-dependent neutrophil phagocytosis (ADNP) and antibody-dependent complement deposition (ADCD), D614G Spike was biotinylated using Sulfo-NHS-LC-LC biotin (Thermo Fisher Scientific) and desalted using Zeba Columns (Thermo Fisher Scientific). Biotinylated antigen was coupled to yellow-green FluoSpheres NeutrAvidin beads (for ADCP and ADNP) or red neutravidin beads (for ADCD) (Invitrogen). To form immune complexes, antigen-coupled beads were incubated with appropriately diluted serum (1:100 for ADCP, 1:50 for ADNP and 1:10 for ADCD) for 2 h at $37\,°C$. Immune complexes were then washed. For ADCP, THP-1 cells (ATCC) were added to plates at a concentration of $2.5 \times 10^4$ cells/mL. Cells were incubated for 16–18 h at $37\,°C$ with the immune complexes and fixed following the incubation. Fluorescence was acquired using an iQue (Intellicyt). For ADNP, leukocytes were isolated from fresh peripheral blood using ACK Lysing Buffer (Thermo Fisher Scientific). Leukocytes were added at a concentration of $5 \times 10^4$ cells/mL. Cells were incubated for 1 h at $37\,°C$ with the

immune complexes. Following the incubation, neutrophils were stained using anti-CD66b Pacblue (Biolegend, cat # 305112, clone G10F5) diluted 1:100 in PBS. Cells were then fixed. Fluorescence of CD66b+ cells was acquired using an iQue (Intellicyt). For ADCP and ADNP, a phago score was calculated using the following formula: (% fluorescent cells*MFI of fluorescent cells)/10,000. For ADCD, lyophilized guinea pig complement (Cedarlane) was diluted in gelatin veronal buffer supplemented with calcium and magnesium. The diluted guinea pig complement was added to immune complexes and plates were incubated at 37 °C for 20 min. Plates were washed with 15 μM EDTA diluted in PBS. Complement was stained using anti-C3 FITC (MP Bio, sku 0855385, cat: 55385, lot 02164) diluted 1:100 in PBS. Fluorescence was determined using an iQue (Intellicyt). For all functional assays, samples were run in duplicate and data are reported as the average of the replicates.

**Antibody-dependent NK cell degranulation.** ELISA plates were coated with 2 μg/mL of Spike protein. Plates were washed and blocked with 5% BSA in PBS. Immune complexes were formed by adding serum diluted 1:25 to plates and incubating plates for 2 h at 37 °C. RosetteSep (STEMCELL Technologies) and a ficoll gradient was used to isolate NK cells from fresh peripheral blood from healthy donors. Isolated NK cells were rested overnight in R10 (see the "Primary cells" section above) with 2 ng/mL of IL-15. NK cells were added to immune complexes at a concentration of $5 \times 10^4$ cells/mL in media supplemented with Brefeldin A (Sigma), anti-CD107a BV605 (Biolegend, Clone H4A3, cat 328634; diluted 1:200) and GolgiStop (BD Biosciences). NK cells were incubated with immune complexes for 5 h at 37 °C. After incubation, cells were stained for surface markers using anti-CD56 PE-Cy7 (BD Biosciences, clone B159, cat 335791, diluted 1:400) and anti- CD3 APC-Cy7 (Biolegend, clone UCHT1, cat 300426, diluted 1:800). Cells were fixed with PermA (Life Technologies), Permeabilized with Perm B (Life Technologies), and stained with anti-MIP1b-BV421 (BD Biosciences, clone D21-1351, cat 562900, diluted 1:800) and anti-IFNg-PE (Biolegend, clone B27, cat 506507, diluted 1:200). The cells were analyzed for fluorescence using an iQue (Intellicyt). NK cells were gates as CD3-/CD56+ and NK activity was determined as the percent of cells positive for CD107a, IFN-g or MIP-1b.

**Multiplexed Luminex assay.** A multiplexed Luminex assay was used to determine the relative concentration of antigen-specific antibody isotype and subclass titer and Fc receptor binding. Carboxylated microsphere was coupled to antigen using EDC and Sulfo-NHS (Thermo Fisher Scientific) to form covalent NHS-ester linkages. To form immune complexes, diluted serum (1:100 for IgG2/3, 1:500 for IgG1, and 1:1000 for FcRs) was mixed with antigen-couple microspheres and incubated overnight at 4 °C shaking at 700 rpm. The following day, plates were washed three times with 0.1% BSA 0.02% Tween-20 in PBS. Antigen-specific antibody isotypes were measured using PE-coupled mouse anti-human antibodies (diluted 1:150) (Mouse Anti-Human IgG1 Fc-PE (Southern Biotech, clone HP6001, cat 9054-09); Mouse Anti-Human IgG2 Fc-PE (Southern Biotech, clone HP6002, cat 9070-09); Mouse Anti-Human IgG3 Hinge-PE (Southern Biotech, clone HP6050, cat 9210-09); Mouse Anti-Human IgA1-PE (Southern Biotech, clone B3506B4, cat 9130-09); Mouse Anti-Human IgM-PE (Southern Biotech, clone UHB, cat 9022-09)). Avi-tagged FcRs (Duke Human Vaccine Institute) was biotinylated using a BirA500 kit (Avidity) and tagged with streptavidin-PE. PE-tagged FcRs were added to immune complexes to determine antigen-specific FcR binding. Fluorescence was acquired using an iQue (Intellicyt) and antigen-specific antibody titer and FcR-binding is reported as Median Fluorescence Intensity (MFI).

**Enzyme-linked immunosorbent assay (ELISA).** To further assess cord:maternal transfer ratios by trimester of vaccination, maternal and umbilical cord blood samples were collected from an additional 52 participants who delivered during the study period ($N = 175$ total maternal-neonatal dyads included for this analysis). Antibodies against the SARS-CoV-2 Spike were quantified using an ELISA. ELISA plates were coated with 500 ng/mL of D614G Spike (kindly provided by Erica Saphire) and incubated for 30 min at room temperature. Plates were washed with washing buffer (0.05% Tween-20. 400 mM NaCl, 50 mM Tris, pH 8.0) and blocked with a 0.1% BSA solution for 30 min at room temperature. Plates were washed, and the sample was added at a dilution of 1:100. Plates were incubated with the sample at 37 °C for 30 min. Plates were washed, and a horseradish peroxidase (HRP)-conjugated goat anti-human IgG antibody (diluted 1:25000) (Bethyl Laboratories, Catalog # A80-219P, lot 20) was added for detection of Spike-specific IgG. Plates were incubated with secondary antibodies for 30 min at room temperature and then washed. TMB was used to develop the ELISA and sulfuric acid was used to stop the ELISA. Signal was read at 450 nm and background was corrected from a reference wavelength of 570 nm.

**SARS-CoV-2 Omicron pseudovirus neutralization assay.** To assess the ability of maternal vaccination to generate neutralizing antibodies, we measured neutralizing activity in a representative subset of 70 maternal participants including participants vaccinated across all three trimesters ($n = 14$ first trimester, 28 second trimester, 28 third trimester vaccine recipients), and with a representation of all three vaccine platforms ($n = 25$ mRNA-1273, $n = 34$ BNT162b2, $n = 11$ Ad26.COV2.S). Omicron spike pseudovirus neutralization assay was performed as previously

described[92]. Briefly, pseudovirus was produced in 293T cells transfected with a lentiviral backbone encoding CMV-Luciferase-IRES-ZsGreen, lentiviral helper plasmids, and the Omicron Spike expression plasmid. Three-fold serial dilutions of serum from 1:12 to 1:8748 were performed, followed by the addition of pseudovirus for 1 h. 293T-ACE2 cells with polybrene were added and incubated at 37 °C for 48 h. Cells were lysed and luciferase expression was quantified using a Spectramax L luminometer (Molecular Devices). Percent neutralization was background subtracted and 50% neutralization titer (NT50) values were calculated, with statistics performed in GraphPad Prism.

**Statistical analysis.** For univariate analysis, statistics were calculated using GraphPad Prism version 8.0. Luminex data and ADCD were $log_{10}$-transformed before analysis. For analysis of differences between vaccines or trimesters, significance was determined by a one-way ANOVA followed by posthoc Benjamini–Hochberg adjustment. For analysis of maternal-cord differences, significance was determined by a Wilcoxon matched-pairs signed-rank test followed by posthoc Benjamini–Hochberg adjustment. Multivariate analysis was performed in R (version 4.0.0). Prior to building the models, data were centered and scaled. LASSO feature selection was performed using the "select_lasso" function in systemseRology R package (v1.0) (https://github.com/LoosC/systemsseRology) to determine significant features. The LASSO tuning parameter was determined by fivefold cross validation. LASSO feature selection was performed 100 times, and features that were chosen 50% of the repetitions were selected to build the model. LASSO-selected features were used to build partial least squares discriminant analysis (PLSDA) or multilevel PLSDA models. Model performance was determined by 5-fold cross validation and significance was evaluated by permutation testing.

**Reporting summary.** Further information on research design is available in the Nature Research Reporting Summary linked to this article.

## Data availability
The Systems Serology data and cord/maternal titer data generated in this study are provided in the Source Data files 1 and 2. Source data are provided with this paper.

## Code availability
No custom code was generated in this manuscript. All code is based on the systemseRology R package (v1.0) (https://github.com/LoosC/systemsseRology).

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

## Acknowledgements

NICHD: 1R01HD100022-01 and 3R01HD100022-02S2 to A.G.E. and 1K12HD103096 to L.L.S., March of Dimes Grant 6-FY20-223 to A.G.E., NIH/NHLBI: K08HL1469630-03 and 3K08HL146963-02S1 to K.J.G., NIAID: R01A1145840-02S1 and 1U19AI167899-01 to G.A., M.A.E., A.G.E., Ragon Institute of MGH, MIT, and Harvard and the MGH ECOR Scholars award to G.A., Nancy Zimmerman, SAMANA Kay MGH Research Scholars award to G.A., an anonymous donor, the Massachusetts Consortium on Pathogen Readiness (MassCPR), the NIH: 3R37AI080289-11S1, R01AI146785, U19AI42790-01, U19AI135995-02, 1U01CA260476-01, and CIVIC5N93019C00052 to G.A., the Gates Foundation Global Health Vaccine Accelerator Platform funding: OPP1146996 and INV-001650 to G.A., and the Musk Foundation. NIDA: Avenir New Innovator Award DP2DA040254 to A.B.B. MGH Transformative Scholars Program to A.B.B.

## Author contributions

C.G.A., L.L.S., K.J.G., G.A., and A.G.E developed the concept, designed the study, and analyzed and interpreted the data. C.G.A., M.S., and A.B.B. performed experiments. C.G.A. performed statistical modeling. C.G.A., L.L.S., and A.G.E. wrote the main paper. S.B., R.M.D., S.D., M.K.K., and C.D.P. collected and prepared samples. L.L.S., C.M., B.O., A.M.B., E.M., M.B., D.C., I.T.G., and A.G.E. recruited and enrolled participants. R.N. and A. Fialkowski assisted with clinical data collection. I.T.G., L.M., A. Fasano, A.B.B., and M.A.E. gave conceptual advice and edited the manuscript. All authors approved of the final manuscript.

## Competing interests

K.J.G. has consulted for Illumina, BillionToOne, and Aetion outside the scope of the submitted work. G.A. is a founder/employee of Seromyx Systems and Leyden Labs. A.G.E. and M.A.E. reported serving as medical advisors for Mirvie, Inc outside the submitted work. A.F. reported serving as a co-founder of and owning stock in Alba Therapeutics and serving on scientific advisory boards for NextCure and Viome outside the submitted work. All other authors declare no competing interests.
