## [Peer review file · Nature Communications]

REVIEWER COMMENTS

Reviewer #1 (Remarks to the Author):

This is a very important study that reveals the quality and efficacy of Ig transfer from the mother to the fetus. The study provides a detailed immunological characterization of the antibody response to the three existing vaccines modalities in the US and its impact on transplacental antibody transfer.

Their finding that anti-Spike antibody titers in umbilical cord blood were 300 higher than maternal titers at delivery when vaccination occurred in the first and second but not at third trimesters supports the concept of early vaccination to provide a better protection to the mother and the fetus.

In addition the demonstration of that the quality of the maternal response has a major impact on the quality of the Ig transfer, has implication to the impact of Sars-cov2 infection during pregnancy. Recommend publication on its present form

Minor comments:

The labels of some of the figures need to be reviewed

- Figure 4 C. Labels on the Y and X axis are difficult to read
- Increase the font on figures 1 and 2 A-C

Reviewer #2 (Remarks to the Author):

The manuscript "Maternal immune response and placental antibody transfer after COVID-19 vaccination across trimester and platforms" submitted by Atyeo et al. presents a study of utmost importance as it gives previously unknown insights into the efficiency of COVID-19 vaccination during pregnancy.

Overall, the study is of very high quality, the conclusions drawn by the authors are almost fully supported by the presented data and the manuscript is written exceptionally well.

I still have comments and suggestions to further improve the manuscript:

1. My most prominent point of criticisms are of methodological nature:

The study measures and relates strength of the IgG response as well as different IgG effector functions. However, a) an overall reduced IgG(1) response is likely to be reflected in reduced effector responses. Maybe the authors performed some kind of normalization (not described in the manuscript) to ensure comparable IgG amounts for functional assays which I would find important?

b) According to the "Methods" different serum dilution were employed for the range of functional assays instead. Why is that?

c) Did you take S-specific IgM into account which might interfere with functional assays (either by blocking antigen-binding sites on beads or by themselves contributing to observed effects)?

d) Why was guinea pig serum used for ADCD assays instead of human serum which would provide much more relevant results?

e) And lastly, why was FcγR1a not included in the different assays even though it could also impact FcR effector functions?

I would greatly appreciate the authors elaborating on these limitations of the study in its current state and preferably extending their data sets accordingly.

2. In figure 1, anti-S1 and S2 IgG responses are evaluated. It would be helpful if the introduction would contain information on the difference and relevance of that differentiation. Can the authors elaborate or speculate on reasons for reduced S2 responses upon As26.COVID.S vaccination?

3. For figure 4B the authors emphasize the reduced ADCD for BTN162b2 recipients. From my perspective this difference is more due to few samples with pronounced reductions rather than a general effect implied by statistics.

4. The identified IgG subclass specific transfer (Fig. 4D-E) is compelling. Has this been observed before for other vaccines or "normal" IgG transfer? What could be the reason for that?

5. In the discussion, the authors state the "... receiving COVID-19 vaccination in the first trimester may therefore optimize benefit for mother and fetus" followed by "by protecting against hospitalization, severe morbidity, and death". I absolutely agree with the first part but find the second part too speculative (although reasonable and likely) in absence of supporting data or literature. Also in the discussion the statement "Although univariate analyses examining responses by trimester did not reveal trimester-specific differences" seems to lack a reference.

6. In fig. 4D-E data sets are presented in orange and blue likely representing mRNA vs. Ad26 vaccine but this information is missing in the figure.

Reviewer #3 (Remarks to the Author):

Key results

- This study evaluates the immune response in 158 pregnant persons by trimester following COVID-19 vaccination with three different SARS CoV-2 vaccines using 2 different vaccine platforms. Applying a systems immunology approach, the authors describe Spike and FcRn binding Ab as well as functional immune responses such as phagocytosis, complement deposition and NK cell activation.

o two mRNA vaccines (Pfizer and Moderna) elicit an enhanced immune response following vaccination, compared to an adenoviral vectored vaccine (Johnson&Johnson).

o Immunization in the second trimester elicited the lowest immune response, while transplacental transfer was highest in first and second trimester

- What is missing:

o the authors omit to evaluate, link, and compare the vaccine-induced neutralizing Ab (nAb) response which to date serves as the primary marker for vaccine efficacy

o real world effectiveness data in pregnant women are available for all three vaccines and need to be linked to allow for any conclusions about the potential role of non-nAB immune response comparisons on vaccine effectiveness in pregnant women and their infants

o more cord blood and infant data are needed to opine on optimal timing of vaccination in pregnancy

Limitations:

- While the study results are novel and of significant scientific interest to the research community, the potential implications of the observed differences in humoral and functional responses on real world clinical outcomes and therefore use of Covid-19 vaccine in pregnant women and their infants are uncertain.

- In order to adequately assess the potential contributory role of cellular immune response functions for comparative vaccine effectiveness, data on vaccine-induced neutralizing antibodies need to be shown in the same study population, and correlated with effectiveness outcomes

- Ad26.COVS vaccine is one of the most widely used vaccine globally and in many geographies the only vaccine recommended and available to pregnant women. As demonstrated in a recent study, high protection against hospitalization and death was achieved by vaccinating pregnant women with mRNA (Pfizer, Moderna) or adenoviral vectored vaccine (Oxford-Astra Zeneca) (Stock S et al, Nature Med 2022).

- The sample size and data are insufficient to draw conclusions on the potential comparative vaccine effectiveness in infants. Earlier vaccination in pregnancy may confer longer coverage during pregnancy

but may not have a differential clinical effect on the infant. In the absence of a correlate of protection it is not clear whether the observed titer differences in the limited number of cord blood samples are of clinical relevance.

- The study data are insufficient to support the authors' conclusions regarding optimal timing of vaccination in pregnancy.

Abstract

- Line 72: While the unit of measure for the transfer of Ab from the maternal to cord blood is the cord:maternal IgG ratio, the measure is generally described as the transplacental transfer ratio, not as the "cord:maternal transfer ratio"

- Line 73: The interpretation of the higher transfer ratios observed in the first and second trimester as "compensation for waning maternal titers" is speculative. A potential saturation of FcR receptors at higher maternal IgG concentrations (=threshold) has been suggested by other experts.

- Line 74: The data are insufficient to support the conclusion of the abstract regarding optimal timing of COVID 19 vaccination in pregnancy, and subsequent protection in infants

- Line 75: Cord blood titers relative to timing of vaccination may be a more appropriate proxy for neonatal antibody protection than transfer ratios

General comments

- The three study groups represent 3 different vaccine candidates, but only 2 different vaccine platforms. There are 28 subjects in the adeno-vectored vaccine group, and 130 subjects in the mRNA vaccine group (comprising Moderna and Pfizer vaccine) which may potentially create a bias (>4.5-fold larger sample size for the mRNA platform) when comparing data across platforms. This should be addressed in the comparative analysis and interpretation of the data.

- In how far are the observed differences between the two platforms unique to pregnancy versus similar to those that have been described in non-pregnant adults?

- What is the IgG Ab subclass profile per vaccine?

- LASSO: which minimal set of features was selected?

- Figure 2C: is a comparison with the mean percentile rank meaningful given the small number the small number of subjects vaccinated in 1st trimester?

- Figure 2D: Are there any study population factors that differ for the Moderna versus Pfizer group?

- Potential clinical meaning (vaccine effectiveness) of enrichment in IgA and IgG2 (correlates with lower transfer ratios) is not clear

- The differences in transfer ratios may be less predictive of clinical protection than the actual cord blood levels – a recent study has shown that although Tdap is associated with higher transfer ratio in late gestation, there was no impact of timing of vaccination on cord blood levels (Clements 2020)

- While IgG1 is generally transferred most efficiently, transfer hierarchies may differ across different populations for the other subclasses.

- In order to adequately assess and compare the immune response profile for each of the three vaccines, the data need to include the currently used primary vaccine immunogenicity marker. i.e. neutralizing Ab and a discussion of the observed immune response differences and their potential correlation with clinical effectiveness of Covid-19 vaccine

- Line 416: Whether neonatal protection is a primary versus a secondary consideration when developing vaccine recommendations in pregnancy is pathogen and disease dependent. For Tdap maternal immunization it is primary, while for COVID-19 it may be secondary. In addition to transfer ratios, conclusions on optimal timing in pregnancy should be informed by predictive immunological and clinical data in the infants, as well as safety data

- Line 438: include reference which did not determine a correlation between high transfer rates and high cord blood titers (Clements 2020).

Reviewer #1 (Remarks to the Author):

This is a very important study that reveals the quality and efficacy of Ig transfer from the mother to the fetus. The study provides a detailed immunological characterization of the antibody response to the three existing vaccines modalities in the US and its impact on transplacental antibody transfer.

Their finding that anti-Spike antibody titers in umbilical cord blood were 300 higher than maternal titers at delivery when vaccination occurred in the first and second but not at third trimesters supports the concept of early vaccination to provide a better protection to the mother and the fetus.

In addition the demonstration of that the quality of the maternal response has a major impact on the quality of the Ig transfer, has implication to the impact of Sars-cov2 infection during pregnancy. Recommend publication on its present form

Minor comments:

The labels of some of the figures need to be reviewed

- Figure 4 C. Labels on the Y and X axis are difficult to read
 - Increase the font on figures 1 and 2 A-C
- Response: We have made the suggested changes.

Reviewer #2 (Remarks to the Author):

The manuscript "Maternal immune response and placental antibody transfer after COVID-19 vaccination across trimester and platforms" submitted by Atyeo et al. presents a study of utmost importance as it gives previously unknown insights into the efficiency of COVID-19 vaccination during pregnancy.

Overall, the study is of very high quality, the conclusions drawn by the authors are almost fully supported by the presented data and the manuscript is written exceptionally well.

I still have comments and suggestions to further improve the manuscript:

1. My most prominent point of criticisms are of methodological nature:

The study measures and relates strength of the IgG response as well as different IgG effector functions. However, a) an overall reduced IgG(1) response is likely to be reflected in reduced effector responses. Maybe the authors performed some kind of normalization (not described in the manuscript) to ensure comparable IgG amounts for functional assays which I would find important?

Response: We thank the reviewer for this suggestion. While we believe the overall effector functions are of greatest clinical relevance after vaccination, rather than a "per IgG" titer-

corrected function, we have performed an analysis correcting the functions for overall titer to isolate functional differences. We have performed a titer correction by dividing the maternal functional responses by IgG1 titer, and demonstrated persistent functional differences between antibodies induced by the Ad26.CoV2.S vaccine compared to the mRNA vaccines. Titer correction did not alter the observed deficit in ADCP, ADNP and CD107a activity of the antibodies generated by Ad26.COV2.S vaccine compared to mRNA vaccines.

This figure has been added to the manuscript as Supplemental Figure 2A.

b) According to the "Methods" different serum dilution were employed for the range of functional assays instead. Why is that?

Response: Before each assay, a dilution test is run to determine the dilution that optimizes signal to background in the assay. Therefore, the selection of dilution is different for each assay. Relevant to this comment and the ones that follow, our approach to these and other component assays that comprise Dr. Galit Alter's systems serology platform has been described extensively in prior publications (selected publications cited here).¹⁻⁹

c) Did you take S-specific IgM into account which might interfere with functional assays (either by blocking antigen-binding sites on beads or by themselves contributing to observed effects)?

Response: It is known that IgM can drive complement deposition,¹⁰ and thus it is likely that some of the observed ADCD activity was driven by IgM. However, given that we did not observe

any differences in the IgM titer between vaccines, we expect that the differences we observed in functional activity were not due to different blocking effects by IgM.

d) Why was guinea pig serum used for ADCD assays instead of human serum which would provide much more relevant results?

Response: Complement has been shown to be highly conserved across species.¹¹ Previous work has shown that guinea pig complement activity in the ADCD assay used here is highly correlated to that of fresh human complement.¹² Previous variation has been attributed in the complement community to variation in the assay itself and the timing of the use of complement that is rapidly inactivated if not used rapidly. The ADCD assay, a GLCP qualified assay, has assessed the optimal use of complement, demonstrating robust correlation with human complement- but critically also providing robust reproducibility and scalability that is not always possible with human serum. For these reasons Guinea Pig complement represents a remarkably close proxy to human complement and offers an important commercial source of a critical reagent.

e) And lastly, why was FcγR1a not included in the different assays even though it could also impact FcR effector functions?

Response: FcγR1 is a high affinity receptor and is likely occupied perpetually by antibodies. Therefore, we did not measure FcγR1 binding since this readout would likely be reflective simply of IgG titer rather than changes in the Fc of IgG.

I would greatly appreciate the authors elaborating on these limitations of the study in its current state and preferably extending their data sets accordingly.

Response: We have elaborated on our methodology in detailed responses above, and have added the titer-corrected antibody functions as a supplemental Figure as described. Our group, as well as numerous other collaborators, have used and reported on the Alter laboratory's systems serology platform in over 30 peer-reviewed publications.

2. In figure 1, anti-S1 and S2 IgG responses are evaluated. It would be helpful if the introduction would contain information on the difference and relevance of that differentiation. Can the authors elaborate or speculate on reasons for reduced S2 responses upon As26.COVID.S vaccination?

Response: The SARS-CoV-2 Spike (S) protein is comprised of two subunits - S1 and S2, which are responsible for host cell receptor attachment and membrane fusion, respectively. Both COVID-19 infection and vaccination generate antibodies against multiple Spike epitopes, including those that bind specifically to S1 and S2 subunits, and there is growing evidence that antibodies against each subunit may contribute distinctly to immune protection.^{13,14} In general, S1, which contains the receptor binding domain (RBD), is thought to be more specific to SARS-CoV-2 and responsible for generating a neutralizing response, whereas S2 is more conserved across human coronaviruses and may play a more active role in memory B-cell and T-cell immunity.¹⁴ In early assessments of COVID-19 infected individuals, blunted S2 serological responses were observed in non-survivors of COVID-19, thought to be evidence of a lack of preexisting, potentially protective coronavirus immunity.¹⁵ Supporting this concept that S2 IgG

generated in response to SARS-CoV-2 reflects an immunological “recall” response, anti-S2 IgG generated by COVID-19 infection and vaccination have been shown to be more cross-reactive against SARS-CoV and MERS-CoV.¹⁶ A recent, comprehensive investigation of the B-cell repertoire before and after vaccination with BNT162b2 elegantly demonstrates that the first vaccine dose generates an early S2-specific “recall” response from memory B cells, purportedly those generated by prior exposure to other coronaviruses, whereas later responses to the second vaccine dose elicit a more specific S1- and RBD-specific response from naive B cells.¹⁷ This response was dramatically boosted by a second dose and delivers potentially neutralizing antibodies against the ancestral strain of SARS-CoV-2. It therefore is also possible that S2 responses require a second “boost” dose to develop a full response; none of the recipients of the Ad26.COV2.S in this study received a second vaccine dose.

The lower S2 responses observed in Ad26.COV2.S recipients compared to mRNA vaccine recipients observed in Figure 1 resonates with other vaccine comparison studies, in which serological and cellular immune responses and overall protection is initially lower after vaccination with Ad26.COV2.S compared to mRNA vaccines.^{18,19} The relative deficiency in S2 response may be driving the observed reduction in overall Spike-specific antibody titers and functions in Ad26.COV2.S vaccine recipients. Differences in S1 responses were not observed, perhaps as the result of high immunogenicity to S1 across all vaccine platforms.

In the process of developing the Ad26.COV2.S vaccine, introduction of stabilizing substitutions including furin cleavage site mutations and two consecutive prolines in the hinge region of S2 in the development of Ad26.COV2.S vaccine increased the ratio of neutralizing to non-neutralizing antibody binding.²⁰ Whether these fundamental differences in antigen design may be responsible for the decreased S2 response observed in Ad26.COV2.S vaccinees in our study is speculative, as equivalent S1 and S2 antibody responses were demonstrated in initial studies of the Ad26.COV2.S vaccine.⁹ This comment resulted in a change to the manuscript—we have included information on what is known about S1 and S2-specific IgG responses (P.8) to support the relevance of probing differences against Spike epitopes, as requested.

3. For figure 4B the authors emphasize the reduced ADCD for BNT162b2 recipients. From my perspective this difference is more due to few samples with pronounced reductions rather than a general effect implied by statistics.

Response: We appreciate the Reviewer’s point and have removed the phrase emphasizing this difference (P.13).

4. The identified IgG subclass specific transfer (Fig. 4D-E) is compelling. Has this been observed before for other vaccines or “normal” IgG transfer? What could be the reason for that?

Response: Our group has shown that in response to vaccination - whether seasonal influenza, Tdap, or mRNA COVID vaccines - preferential transfer of highly functional, NK cell-activating FcR-binding antibodies from maternal to cord blood occurs (i.e. “transplacental transfer”).^{1,5,21} In a recent publication, we demonstrated increased S2 and RBD-specific FcR2a, 2b and 3a binding in cord blood compared to maternal blood following COVID-19 mRNA vaccination in a cohort of only third trimester mRNA vaccine recipients, indicative of transfer of highly functional antibodies.²¹ We have also shown that compared to influenza or Tdap vaccine-derived antibodies, transfer of anti-SARS-CoV-2 antibodies in the setting of COVID-19 infection is

impaired, due in part to altered SARS-CoV-2 antibody glycosylation profiles that interfere with effective transfer. Even in the setting of impaired overall transfer of anti-SARS-CoV-2 antibodies, we observed selective transfer of functional NK-activating acfucosylated antibodies to the cord.¹ This work has shed significant light on the capability of the placenta to populate the newborn with highly active, protective antibodies - whether in the setting of vaccination or infection. Our results presented in Figure 4 are consistent with this prior work showing unique placental sieving mechanisms for populating the newborn with highly functional IgG both in the setting of vaccination (Tdap, COVID-19) and natural infection (SARS-CoV-2). In response to this comment, we have added text to place these results in the context of the prior literature, as reviewed above (P.14).

5. In the discussion, the authors state the "... receiving COVID-19 vaccination in the first trimester may therefore optimize benefit for mother and fetus" followed by "by protecting against hospitalization, severe morbidity, and death". I absolutely agree with the first part but find the second part too speculative (although reasonable and likely) in absence of supporting data or literature. Also in the discussion the statement "Although univariate analyses examining responses by trimester did not reveal trimester-specific differences" seems to lack a reference. Response: We appreciate the opportunity to clarify this point. The second portion of the statement is not speculative. Large epidemiologic studies have established the efficacy of COVID-19 vaccination in preventing against SARS-CoV-2 infection, hospitalization, severe morbidity and death in pregnant individuals, and these results have been reported in the literature.²²⁻²⁵ Protecting the pregnant individual against the increased severe maternal morbidity and mortality associated with COVID-19 in pregnancy^{26,27} remains the primary objective of COVID-19 vaccination in pregnancy, per the Centers for Disease Control and Prevention, American College of Obstetricians and Gynecologists, and the Society for Maternal-Fetal Medicine.²⁸⁻³⁰ We have added five references (Stock et al. Nat Med 2022, Thelier et al. AJOG MFM 2021, Morgan et al. Obstet Gynecol 2022, Dagan et al. Nature Medicine 2021, Engjom et al., Lancet Regional Health, 2022) to support the statement that COVID-19 vaccines protect pregnant people against severe or critical COVID-19 illness and hospitalization and have rephrased the statement as such (P. 4 and P. 21). The statement "Although univariate analyses examining responses by trimester did not reveal trimester-specific differences" refers to the data presented in Figure 2A-B.

6. In fig. 4D-E data sets are presented in orange and blue likely representing mRNA vs. Ad26 vaccine but this information is missing in the figure.

Response: We believe the Reviewer is referring to Figure 5 D and E. We appreciate the need for clarification in this regard. The colors reference trimester of vaccination, with orange 1st trimester vaccination, blue 2nd, and pink 3rd trimester vaccination, and this legend appears in a small box in the figure between panels A and B and panels C,D,E. This legend box has been enlarged slightly in the revised figure as we agree it was easily missed prior. The information about which color refers to which trimester of vaccination is also presented in the legend for figure 5A and in the figure labeling for Figures B, C, and E but we have also added additional text about the colors representing trimesters of vaccination to the legend text for 5D and 5E, for clarity.

Reviewer #3 (Remarks to the Author):

Key results

- This study evaluates the immune response in 158 pregnant persons by trimester following COVID-19 vaccination with three different SARS CoV-2 vaccines using 2 different vaccine platforms. Applying a systems immunology approach, the authors describe Spike and FcRn binding Ab as well as functional immune responses such as phagocytosis, complement deposition and NK cell activation.

o two mRNA vaccines (Pfizer and Moderna) elicit an enhanced immune response following vaccination, compared to an adenoviral vectored vaccine (Johnson&Johnson).

o Immunization in the second trimester elicited the lowest immune response, while transplacental transfer was highest in first and second trimester

- What is missing:

o the authors omit to evaluate, link, and compare the vaccine-induced neutralizing Ab (nAb) response which to date serves as the primary marker for vaccine efficacy

Response: We agree that neutralizing antibodies (nAbs) are useful as a marker in comparing vaccine efficacy across vaccines or vaccine platforms, as nAb titers correlate with clinical protection. Multiple correlate-of-protection studies in non-pregnant cohorts have demonstrated that higher antibody titers (both binding and neutralizing) are associated with a decreased risk of subsequent symptomatic infection.³¹⁻³³ Prior analyses have demonstrated a high degree of correlation between both mean peak neutralizing titers and vaccine-induced anti-S IgG binding antibodies and decreased risk of infection,^{32,33} and thus we feel our conclusions suggesting a correlation between anti-Spike antibody titers and protection from severe disease are substantiated by this literature.

Our group and others have previously demonstrated the induction of neutralizing antibodies to ancestral Spike after mRNA vaccination in pregnancy.³⁴⁻³⁶ As the Omicron variant is the primary circulating strain in the United States (<https://covid.cdc.gov/covid-data-tracker/#variant-proportions>), neutralizing titers against Omicron are the most relevant to report on at this time. Several studies have suggested that vaccine-induced antibody neutralization against the Omicron variant is significantly reduced relative to neutralization against ancestral Spike, primarily in nonpregnant populations with limited data available in pregnancy.³⁷⁻⁴¹ We have now included data on neutralization against Omicron in Figure S2B and S2C. We observed that there is a limited induction of neutralizing antibodies against the Omicron variant, with 39% of maternal sera demonstrating neutralizing activity, consistent with previous studies in which 6-24% of sera from mRNA vaccine recipients had neutralizing activity against Omicron.^{38,39,41} These data point to the need to understand other immunological features that can provide protection against disease in pregnant women, such as antibody Fc function, which have been shown to be maintained in pregnant populations against Omicron.⁴²

These points have been added to the Results (P.9) and Discussion (P. 18)

o real world effectiveness data in pregnant women are available for all three vaccines and need to be linked to allow for any conclusions about the potential role of non-nAB immune response comparisons on vaccine effectiveness in pregnant women and their infants

Response: We have included updated references to these population level data demonstrating the effectiveness of all three COVID-19 vaccines in preventing infection and severe disease in pregnant people (P.4). We have also referenced recently-published data from the CDC demonstrating that maternal mRNA vaccination is 61% effective in preventing newborn hospitalization from COVID-19 in the first 6 months of life.⁴³

o more cord blood and infant data are needed to opine on optimal timing of vaccination in pregnancy

This manuscript contains data not only on the 123 dyads in whom systems serology was performed, but also an augmented cohort containing an additional 52 dyads for a total of 175 maternal blood:cord blood dyads with which to answer questions about optimal timing of vaccination in pregnancy. This represents a relatively large cohort for the literature at this time. In a recent publication of data from a cohort of 77 infants whose mothers received a COVID-19 mRNA vaccine at 20-32 weeks of pregnancy, we demonstrate that anti-Spike IgG is present in 98% of infants at 2 months of life, and in 57% at 6 months of life.⁴⁴ Additionally we demonstrate that infant titers at 2 months were highly correlated with both maternal and cord titers at delivery. We have also demonstrated in our other prior work²¹ and in the analyses presented in Figure 4 that unique placental sieving mechanisms populate the infant with highly functional antibodies. Based on these data, and population level data from the CDC demonstrating that maternal mRNA vaccines are 61% effective in preventing newborn hospitalization from COVID-19 infection in the first 6 months of life,⁴³ we feel it is reasonable to suggest that the presence of anti-Spike IgG could be a correlate of protection against COVID-19 infection and/or severe disease in the newborn, although we agree that large-scale, observational cohort studies would be needed to accurately quantify the degree of protection associated with specific titers.

Limitations:

- While the study results are novel and of significant scientific interest to the research community, the potential implications of the observed differences in humoral and functional responses on real world clinical outcomes and therefore use of Covid-19 vaccine in pregnant women and their infants are uncertain.

Response: Our study was not designed to examine COVID-19 infection rates in vaccinated pregnant women or their newborns, and was not powered to show differences between groups for such outcomes, which are more appropriately examined with large population-level studies. Such studies have been performed and we appreciate the point that these data should be highlighted—additional sentences with references have been added to emphasize that large epidemiologic studies have demonstrated efficacy of the COVID-19 vaccines in pregnancy to reduce infection and prevent severe disease and death in pregnant individuals and prevent newborn COVID-19 hospitalizations (P. 4). Such studies are not able, however, to help elucidate antibody correlates of protection by performing this type of detailed antibody profiling. Both types of studies in combination are needed to best understand how antibody characteristics and function might provide protection against infection and severe disease.

However, while previous studies investigating vaccine responses in pregnancy have focused primarily on unidimensional measures such as antibody titer, the strength of the systems serology approach used in our study is that it allows for assessment of cellular functional responses beyond titer alone, demonstrating antibody functionality as a correlate for protection. The degree to which observed differences in functional response directly relate to differences in clinical effectiveness is not known and cannot be answered by one study; data such as those presented here will always need to be interpreted in combination with large, population-level studies that themselves are unable to perform this type of detailed profiling.

- In order to adequately assess the potential contributory role of cellular immune response functions for comparative vaccine effectiveness, data on vaccine-induced neutralizing antibodies need to be shown in the same study population, and correlated with effectiveness outcomes

Response: We have now included data on neutralizing antibody (nAb) responses across vaccine platforms from the cohort as requested. For full details, please see our response to the point above requesting nAb data. As stated in the response to the prior point, it is not the purpose of this study to report on clinical vaccine efficacy, our study focused on deep antibody profiling across vaccine platforms and trimester of vaccination. Studies reporting on vaccine efficacy require many hundreds to thousands of patients, depending on the incidence of severe/reportable disease, and have a different design and purpose.

- Ad26.COV2.S vaccine is one of the most widely used vaccine globally and in many geographies the only vaccine recommended and available to pregnant women. As demonstrated in a recent study, high protection against hospitalization and death was achieved by vaccinating pregnant women with mRNA (Pfizer, Moderna) or adenoviral vectored vaccine (Oxford-Astra Zeneca) (Stock S et al, Nature Med 2022).

Response: We appreciate this point and agree with the importance of studying and reporting on the adenoviral vectored vaccine; we have added this reference in our manuscript (reference 9).

- The sample size and data are insufficient to draw conclusions on the potential comparative vaccine effectiveness in infants. Earlier vaccination in pregnancy may confer longer coverage during pregnancy but may not have a differential clinical effect on the infant. In the absence of a correlate of protection it is not clear whether the observed titer differences in the limited number of cord blood samples are of clinical relevance.

Response: We agree that these data alone are not sufficient to draw conclusions on potential comparative vaccine effectiveness in infants, and this was not the purpose of the experiments reported here. However, our results can be placed in the context of available literature and provide important complementary data that can add to knowledge about optimal timing of vaccination in pregnancy to protect both members of the maternal-fetal dyad. Since the time of submission, two papers have been published that offer key data about maternal vaccination and infant protection that complement our data reported here, and our Discussion has been modified to include these (P.21). Work by our own group has shown that anti-Spike titers in the infant at 2 months of age following maternal COVID-19 vaccination are directly correlated with maternal/cord titers at birth⁴⁴ - thus it is reasonable to conclude that cord titers are related to

infant titers. Moreover, recent population-level data from the CDC demonstrated that maternal vaccination in pregnancy was overall associated with a 61% reduction in infant hospitalization for COVID-19 in the first 6 months of life, with greater vaccine efficacy in preventing hospitalization noted for vaccination after 20 weeks gestation.⁴³ Taken together, these two studies provide powerful evidence that vaccination in pregnancy results in detectable infant titers up to 6 months of age in a majority of infants, and can confer clinically-meaningful protection. However, given the potential for severe maternal morbidity and mortality associated with COVID-19 in pregnancy,^{26,27,45,46} the established ability of the COVID-19 vaccines to prevent severe maternal morbidity and mortality,^{23-25,47} and the recently-established associations between COVID-19 in pregnancy and adverse pregnancy outcomes including preterm birth, hypertensive disorders of pregnancy, and stillbirth,⁴⁸⁻⁵⁰ optimal protection of the maternal-fetal dyad must consider more than just cord titers at birth and subsequent infant titers and protection from disease. Given the substantial potential maternal benefit against severe disease incurred by earlier vaccination in pregnancy (and neonatal benefit through protection against spontaneous and iatrogenic preterm birth and stillbirth), the comparable (or potentially superior) maternal immune response we observed with early vaccination in pregnancy, and the placental biology that drives functional titers into the umbilical cord in women vaccinated remote from delivery (as evidenced by the higher transfer ratios we note for first trimester vaccination), we argue that the strategy most beneficial to the maternal-newborn dyad is early vaccination.

- The study data are insufficient to support the authors' conclusions regarding optimal timing of vaccination in pregnancy.

Response: See our response to previous comment. We have modified the Discussion to place our results in the context of additional studies that have been published since this manuscript was submitted and to acknowledge the limitations of our study (P.21). While this study did not measure clinical effectiveness and such measures were not the goal of our study, taken together with other population-level vaccine efficacy data and infant protection data, our data provide key information to complement and augment the available literature. Namely, we demonstrate that first trimester vaccination generates comparable (if not superior) maternal immunogenicity to vaccination in later trimesters, and we demonstrate efficient transfer of maternal IgG across the placenta in both first and second trimester vaccination. These data therefore represent a critical piece of the puzzle and an important adjunct to other studies examining vaccine efficacy to protect against both severe maternal and infant disease.

Abstract

- Line 72: While the unit of measure for the transfer of Ab from the maternal to cord blood is the cord:maternal IgG ratio, the measure is generally described as the transplacental transfer ratio, not as the "cord:maternal transfer ratio"

Response: We have made the suggested change to improve clarity of this statement.

- Line 73: The interpretation of the higher transfer ratios observed in the first and second trimester as "compensation for waning maternal titers" is speculative. A potential saturation of FcR receptors at higher maternal IgG concentrations (=threshold) has been suggested by other experts.

Response: We have softened the language (P.3, P.16) to emphasize that this is a potential mechanism suggested by our observations, and added reference to the possibility of saturation of Fc-receptors with higher maternal IgG concentrations.

- Line 74: The data are insufficient to support the conclusion of the abstract regarding optimal timing of COVID 19 vaccination in pregnancy, and subsequent protection in infants

Response: We have softened language throughout to acknowledge what conclusions the experiments presented here can support. However, the data presented here support that maternal antibody responses are strong following vaccination early in pregnancy, and we argue that vaccinating early in pregnancy would therefore afford the greatest protection for maternal benefit, which is the primary indication for COVID-19 vaccination. Protection of the dyad is about more than just infant protection, and if vaccination earlier in pregnancy provides the longest duration of protection for the mother, this in turn also protects the fetus against pregnancy complications that are strongly associated with preterm birth or stillbirth. Thus, optimizing protection for the maternal-fetal dyad involves considerations beyond cord or infant titer.

- Line 75: Cord blood titers relative to timing of vaccination may be a more appropriate proxy for neonatal antibody protection than transfer ratios.

Response: We appreciate this point and for this reason we had provided information on cord blood anti-Spike titers at delivery (Figure 5C) in addition to transfer ratios (Figure 5B) by trimester of vaccination. We do find, however, that transfer ratios are informative in that they reflect the ability of the placenta to efficiently transfer maternal IgG with vaccination across all trimesters of pregnancy, with potential higher efficiency noted with first and second trimester vaccination relative to third, as per responses above.

General comments

- The three study groups represent 3 different vaccine candidates, but only 2 different vaccine platforms. There are 28 subjects in the adeno-vectored vaccine group, and 130 subjects in the mRNA vaccine group (comprising Moderna and Pfizer vaccine) which may potentially create a bias (>4.5-fold larger sample size for the mRNA platform) when comparing data across platforms. This should be addressed in the comparative analysis and interpretation of the data.

- In how far are the observed differences between the two platforms unique to pregnancy versus similar to those that have been described in non-pregnant adults?

Response: There are limited data comparing the differences across vaccine platforms in non-pregnant adults. Serological and cellular immune responses and overall protection appears to be lower in Ad26.COVID.S vaccine recipients compared to mRNA recipients,^{18,19} although the durability of the Ad26.COVID.S response over time may be superior to the mRNA vaccines.¹⁹ Unfortunately due to the association noted between the adenoviral vectored vaccine and vaccine induced thrombotic thrombocytopenia/venous sinus thrombosis in women of reproductive age, uptake of the Ad26.CoV2.S vaccine was quite limited in the United States. Thus, the 28 subjects in this group were all the subjects available for study, given that the Ad26.CoV2.S vaccine only had real-world use by pregnant patients in the U.S. for a short period

of time. To our knowledge, this number of Ad26.CoV2.S recipients in pregnancy is the largest number reported in the literature to date.

- What is the IgG Ab subclass profile per vaccine?

Response: We agree that understanding the subclass profile is vital to understanding the immunogenicity of the vaccines. We note that IgG2 and IgG3 titer by vaccine are in supplemental figure 1 (for maternal response) and supplemental figure 3 (for cord titer).

- LASSO: which minimal set of features was selected?

Response: The set of features selected are present in Figure 1E (for maternal titer) and Figure 3E (cord titers). We have ensured these results are clearly referenced in the manuscript.

- Figure 2C: is a comparison with the mean percentile rank meaningful given the small number of subjects vaccinated in 1st trimester?

Response: We appreciate the reviewer's point about mean percentile rank. Therefore, we have toned the language as follows (P.10):

“This analysis revealed that both first and third trimester vaccination drove a higher, albeit not significant, functional antibody response compared to second trimester vaccination, marked by both higher FcR-binding and more functional antibodies as indicated by enhanced ADCD, ADNP, ADCP, and ADNKA responses.”

- Figure 2D: Are there any study population factors that differ for the Moderna versus Pfizer group?

Response: Table 1 presents demographic and clinical characteristics of vaccine recipients by vaccine platform. In this cohort, there was no significant differences noted in maternal age, gravidity, parity, race, ethnicity, insurance status, BMI, presence of an autoimmune condition, known prior SARS-CoV-2 infection, or time elapsed between vaccination and sample collection, between mRNA-1273 and BNT162b2 vaccine recipients.

- Potential clinical meaning (vaccine effectiveness) of enrichment in IgA and IgG2 (correlates with lower transfer ratios) is not clear

Response: We describe these findings to highlight potential differences in vaccine-induced responses, however we do not assert that taken in isolation these findings have immediate clinical applications. Please see above answers for discussion of what our study can elucidate, compared to population-level studies that examine clinical vaccine efficacy.

- The differences in transfer ratios may be less predictive of clinical protection than the actual cord blood levels – a recent study has shown that although Tdap is associated with higher transfer ratio in late gestation, there was no impact of timing of vaccination on cord blood levels (Clements 2020)

Response: We agree with this point, and provide information on absolute cord blood anti-Spike titers at delivery (Figure 5C) in addition to transfer ratios (Figure 5B) by trimester of vaccination for the reason that cord titers are likely more reflective of neonatal protection, whereas transfer

ratios provide a metric of transplacental transfer that has been widely used in vaccine literature to compare the effectiveness of placental IgG transfer by vaccine or pathogen within the same individual or across studies.

- While IgG1 is generally transferred most efficiently, transfer hierarchies may differ across different populations for the other subclasses.

Response: We agree with the reviewer that IgG2/IgG3 are important for protection and transfer at different rates than IgG1 due to different affinities for FcRn. We note that these isotypes were included in the multilevel PLSDAs in Figure 4 and are selected as features that separate cord/maternal blood.

- In order to adequately assess and compare the immune response profile for each of the three vaccines, the data need to include the currently used primary vaccine immunogenicity marker. i.e. neutralizing Ab and a discussion of the observed immune response differences and their potential correlation with clinical effectiveness of Covid-19 vaccine

Response: Please see prior responses to previous Reviewer comments that have addressed these points.

- Line 416: Whether neonatal protection is a primary versus a secondary consideration when developing vaccine recommendations in pregnancy is pathogen and disease dependent. For Tdap maternal immunization it is primary, while for COVID-19 it may be secondary. In addition to transfer ratios, conclusions on optimal timing in pregnancy should be informed by predictive immunological and clinical data in the infants, as well as safety data

Response: We agree that for COVID-19 vaccines, maternal immunization is primary, however given that COVID vaccines are unlikely to be available for infants less than 6 months of age, there is significant interest in what factors might optimize neonatal protection. As we have previously noted in prior responses, we have added reference to recent CDC data on the effectiveness of maternal vaccination (61%) in protecting newborns from hospitalization due to COVID-19 infection in this age group, as well as information from our own group demonstrating the presence of vaccine-induced anti-Spike titers in 98% of infants at 2 months of age, and 57% infants at 6 months of age.⁴⁴

- Line 438: include reference which did not determine a correlation between high transfer rates and high cord blood titers (Clements 2020).

Response: We thank the Reviewer for this point. This reference has been added to the discussion on timing of Tdap administration and antibody transfer (reference 76).

References:

1. Atyeo C, Pullen KM, Bordt EA, et al. Compromised SARS-CoV-2-specific placental antibody transfer. *Cell*. 2021;184(3):628-642.e10.
2. Bartsch YC, Wang C, Zohar T, et al. Humoral signatures of protective and pathological SARS-CoV-2 infection in children. *Nat Med*. 2021;27(3):454-462.

3. Zohar T, Loos C, Fischinger S, et al. Compromised Humoral Functional Evolution Tracks with SARS-CoV-2 Mortality. *Cell*. 2020;183(6):1508-1519.e12.
4. Lai JI, Licht AF, Dugast AS, et al. Divergent antibody subclass and specificity profiles but not protective HLA-B alleles are associated with variable antibody effector function among HIV-1 controllers. *J Virol*. 2014;88(5):2799-2809.
5. Jennewein MF, Goldfarb I, Dolatshahi S, et al. Fc Glycan-Mediated Regulation of Placental Antibody Transfer. *Cell*. 2019;178(1):202–215.e14.
6. Boudreau CM, Yu WH, Suscovich TJ, Talbot HK, Edwards KM, Alter G. Selective induction of antibody effector functional responses using MF59-adjuvanted vaccination. *J Clin Invest*. 2020;130(2):662-672.
7. Jennewein MF, Mabuka J, Papia CL, et al. Tracking the Trajectory of Functional Humoral Immune Responses Following Acute HIV Infection. *Front Immunol*. 2020;11:1744.
8. Chung AW, Alter G. Systems serology: profiling vaccine induced humoral immunity against HIV. *Retrovirology*. 2017;14(1):57.
9. Stephenson KE, Le Gars M, Sadoff J, et al. Immunogenicity of the Ad26.COVS.2 Vaccine for COVID-19. *JAMA*. 2021;325(15):1535-1544.
10. Janeway CA Jr, Travers P, Walport M, Shlomchik MJ. *The Distribution and Functions of Immunoglobulin Isotypes*. Garland Science; 2001.
11. Nonaka M, Kimura A. Genomic view of the evolution of the complement system. *Immunogenetics*. 2006;58(9):701-713.
12. Fischinger S, Fallon JK, Michell AR, et al. A high-throughput, bead-based, antigen-specific assay to assess the ability of antibodies to induce complement activation. *J Immunol Methods*. 2019;473:112630.
13. Voss WN, Hou YJ, Johnson NV, et al. Prevalent, protective, and convergent IgG recognition of SARS-CoV-2 non-RBD spike epitopes. *Science*. 2021;372(6546):1108-1112.
14. Shah P, Canziani GA, Carter EP, Chaiken I. The Case for S2: The Potential Benefits of the S2 Subunit of the SARS-CoV-2 Spike Protein as an Immunogen in Fighting the COVID-19 Pandemic. *Front Immunol*. 2021;12:637651.
15. Kaplonek P, Wang C, Bartsch Y, et al. Early cross-coronavirus reactive signatures of humoral immunity against COVID-19. *Sci Immunol*. 2021;6(64):eabj2901.
16. Grobden M, van der Straten K, Brouwer PJ, et al. Cross-reactive antibodies after SARS-CoV-2 infection and vaccination. *Elife*. 2021;10. doi:10.7554/eLife.70330
17. Brewer RC, Ramadoss NS, Lahey LJ, Jahanbani S, Robinson WH, Lanz TV. BNT162b2 vaccine induces divergent B cell responses to SARS-CoV-2 S1 and S2. *Nat Immunol*. 2022;23(1):33-39.
18. Naranbhai V, Garcia-Beltran WF, Chang CC, et al. Comparative immunogenicity and effectiveness of mRNA-1273, BNT162b2 and Ad26.COVS.2 COVID-19 vaccines. *J Infect Dis*. Published online December 9, 2021. doi:10.1093/infdis/jiab593

19. Collier ARY, Yu J, McMahan K, et al. Differential Kinetics of Immune Responses Elicited by Covid-19 Vaccines. *N Engl J Med*. 2021;385(21):2010-2012.
20. Bos R, Rutten L, van der Lubbe JEM, et al. Ad26 vector-based COVID-19 vaccine encoding a prefusion-stabilized SARS-CoV-2 Spike immunogen induces potent humoral and cellular immune responses. *NPJ Vaccines*. 2020;5:91.
21. Atyeo C, DeRiso EA, Davis C, et al. COVID-19 mRNA vaccines drive differential antibody Fc-functional profiles in pregnant, lactating, and non-pregnant women. *Sci Transl Med*. Published online October 19, 2021:eabi8631.
22. Goldshtein I, Nevo D, Steinberg DM, et al. Association Between BNT162b2 Vaccination and Incidence of SARS-CoV-2 Infection in Pregnant Women. *JAMA*. 2021;326(8):728-735.
23. Dagan N, Barda N, Biron-Shental T, et al. Effectiveness of the BNT162b2 mRNA COVID-19 vaccine in pregnancy. *Nat Med*. 2021;27(10):1693-1695.
24. Theiler RN, Wick M, Mehta R, Weaver AL, Virk A, Swift M. Pregnancy and birth outcomes after SARS-CoV-2 vaccination in pregnancy. *Am J Obstet Gynecol MFM*. 2021;3(6):100467.
25. Morgan JA, Biggio JR Jr, Martin JK, et al. Maternal Outcomes After Severe Acute Respiratory Syndrome Coronavirus 2 (SARS-CoV-2) Infection in Vaccinated Compared With Unvaccinated Pregnant Patients. *Obstet Gynecol*. 2022;139(1):107-109.
26. Zambrano LD, Ellington S, Strid P, et al. Update: Characteristics of Symptomatic Women of Reproductive Age with Laboratory-Confirmed SARS-CoV-2 Infection by Pregnancy Status - United States, January 22-October 3, 2020. *MMWR Morb Mortal Wkly Rep*. 2020;69(44):1641-1647.
27. Delahoy MJ, Whitaker M, O'Halloran A, et al. Characteristics and Maternal and Birth Outcomes of Hospitalized Pregnant Women with Laboratory-Confirmed COVID-19 - COVID-NET, 13 States, March 1-August 22, 2020. *MMWR Morb Mortal Wkly Rep*. 2020;69(38):1347-1354.
28. CDC. COVID-19 Vaccines While Pregnant or Breastfeeding. Centers for Disease Control and Prevention. Published February 18, 2022. Accessed February 28, 2022. <https://www.cdc.gov/coronavirus/2019-ncov/vaccines/recommendations/pregnancy.html>
29. ACOG Practice Advisory. COVID-19 Vaccination Considerations for Obstetric–Gynecologic Care. Published July 30, 2021. Accessed September 3, 2021. <https://www.acog.org/clinical/clinical-guidance/practice-advisory/articles/2020/12/covid-19-vaccination-considerations-for-obstetric-gynecologic-care>
30. Society for Maternal-Fetal Medicine. Society for Maternal-Fetal Medicine (SMFM) Statement: SARS-CoV-2 Vaccination in Pregnancy. Published December 1, 2020. Accessed April 28, 2021. [https://s3.amazonaws.com/cdn.smfm.org/media/2591/SMFM_Vaccine_Statement_12-1-20_\(final\).pdf](https://s3.amazonaws.com/cdn.smfm.org/media/2591/SMFM_Vaccine_Statement_12-1-20_(final).pdf)
31. CDC. Science Brief: SARS-CoV-2 Infection-induced and Vaccine-induced Immunity. Centers for Disease Control and Prevention. Published October 30, 2021. Accessed

February 1, 2022. <https://www.cdc.gov/coronavirus/2019-ncov/science/science-briefs/vaccine-induced-immunity.html>

32. Khoury DS, Cromer D, Reynaldi A, et al. Neutralizing antibody levels are highly predictive of immune protection from symptomatic SARS-CoV-2 infection. *Nat Med*. 2021;27(7):1205-1211.
33. Earle KA, Ambrosino DM, Fiore-Gartland A, et al. Evidence for antibody as a protective correlate for COVID-19 vaccines. *Vaccine*. 2021;39(32):4423-4428.
34. Gray KJ, Bordt EA, Atyeo C, et al. COVID-19 vaccine response in pregnant and lactating women: a cohort study. *Am J Obstet Gynecol*. Published online March 24, 2021. doi:10.1016/j.ajog.2021.03.023
35. Collier ARY, McMahan K, Yu J, et al. Immunogenicity of COVID-19 mRNA Vaccines in Pregnant and Lactating Women. *JAMA*. 2021;325(23):2370-2380.
36. Prabhu M, Murphy EA, Sukhu AC, et al. Antibody Response to Coronavirus Disease 2019 (COVID-19) Messenger RNA Vaccination in Pregnant Women and Transplacental Passage Into Cord Blood. *Obstet Gynecol*. Published online April 28, 2021. doi:10.1097/AOG.0000000000004438
37. Sievers BL, Chakraborty S, Xue Y, et al. Antibodies elicited by SARS-CoV-2 infection or mRNA vaccines have reduced neutralizing activity against Beta and Omicron pseudoviruses. *Sci Transl Med*. Published online January 13, 2022:eabn7842.
38. Lu L, Mok BWY, Chen LL, et al. Neutralization of SARS-CoV-2 Omicron variant by sera from BNT162b2 or Coronavac vaccine recipients. *Clin Infect Dis*. Published online December 16, 2021. doi:10.1093/cid/ciab1041
39. Planas D, Saunders N, Maes P, et al. Considerable escape of SARS-CoV-2 Omicron to antibody neutralization. *Nature*. Published online December 23, 2021. doi:10.1038/s41586-021-04389-z
40. Cele S, Jackson L, Khoury DS, et al. Omicron extensively but incompletely escapes Pfizer BNT162b2 neutralization. *Nature*. Published online December 23, 2021. doi:10.1038/s41586-021-04387-1
41. Cheng SMS, Mok CKP, Leung YWY, et al. Neutralizing antibodies against the SARS-CoV-2 Omicron variant BA.1 following homologous and heterologous CoronaVac or BNT162b2 vaccination. *Nat Med*. Published online January 20, 2022. doi:10.1038/s41591-022-01704-7
42. Bartsch Y, Atyeo C, Kang J, Gray KJ, Edlow AG, Alter G. Preserved recognition of Omicron Spike following COVID-19 mRNA vaccination in pregnancy. *bioRxiv*. Published online January 2, 2022. doi:10.1101/2022.01.01.22268615
43. Halasa NB, Olson SM, Staat MA, et al. Effectiveness of Maternal Vaccination with mRNA COVID-19 Vaccine During Pregnancy Against COVID-19-Associated Hospitalization in Infants Aged <6 Months - 17 States, July 2021-January 2022. *MMWR Morb Mortal Wkly Rep*. 2022;71(7):264-270.

44. Shook LL, Atyeo CG, Yonker LM, et al. Durability of Anti-Spike Antibodies in Infants After Maternal COVID-19 Vaccination or Natural Infection. *JAMA*. Published online February 7, 2022. doi:10.1001/jama.2022.1206
45. Villar J, Ariff S, Gunier RB, et al. Maternal and Neonatal Morbidity and Mortality Among Pregnant Women With and Without COVID-19 Infection: The INTERCOVID Multinational Cohort Study. *JAMA Pediatr*. 2021;175(8):817-826.
46. Khan DSA, Pirzada AN, Ali A, Salam RA, Das JK, Lassi ZS. The Differences in Clinical Presentation, Management, and Prognosis of Laboratory-Confirmed COVID-19 between Pregnant and Non-Pregnant Women: A Systematic Review and Meta-Analysis. *Int J Environ Res Public Health*. 2021;18(11). doi:10.3390/ijerph18115613
47. Engjom H, van den Akker T, Aabakke A, et al. Severe COVID-19 in pregnancy is almost exclusively limited to unvaccinated women - time for policies to change. *Lancet Reg Health Eur*. 2022;13:100313.
48. Metz TD, Clifton RG, Hughes BL, et al. Association of SARS-CoV-2 Infection With Serious Maternal Morbidity and Mortality From Obstetric Complications. *JAMA*. 2022;327(8):748-759.
49. Stock SJ, Carruthers J, Calvert C, et al. SARS-CoV-2 infection and COVID-19 vaccination rates in pregnant women in Scotland. *Nat Med*. Published online January 13, 2022. doi:10.1038/s41591-021-01666-2
50. Schwartz MD MS Hyg D, Avvad-Portari E, Babál P, et al. Placental Tissue Destruction and Insufficiency from COVID-19 Causes Stillbirth and Neonatal Death from Hypoxic-Ischemic Injury: A Study of 68 Cases with SARS-CoV-2 Placentitis from 12 Countries. *Arch Pathol Lab Med*. Published online February 10, 2022. doi:10.5858/arpa.202-0029-SA

REVIEWER COMMENTS

Reviewer #2 (Remarks to the Author):

I thank the authors for their detailed response to my comments and appreciate including additional information in the manuscript. My points of concern have been fully addressed.

Reviewer #3 (Remarks to the Author):

The study provides an interesting characterization of immune responses by trimester following vaccination in pregnancy by evaluating a set of immunologic biomarkers using systems serology methodology. The study design and data do not support conclusions and recommendations regarding correlates or surrogates of protection and optimal timing of vaccination in pregnancy in the absence of correlating these with clinical efficacy in mothers and babies and using validated assays.

In this regard, speculative statements need to be removed accordingly and the descriptive nature of the data should be clarified.

Relevant references:

Sadarangani M, Nature Reviews 2021

Plotkin S, Clin Vacc Immunology 2010

REVIEWER COMMENTS

Reviewer #2 (Remarks to the Author):

I thank the authors for their detailed response to my comments and appreciate including additional information in the manuscript. My points of concern have been fully addressed.

Response: We appreciate the Reviewer's efforts and comments.

Reviewer #3 (Remarks to the Author):

The study provides an interesting characterization of immune responses by trimester following vaccination in pregnancy by evaluating a set of immunologic biomarkers using systems serology methodology. The study design and data do not support conclusions and recommendations regarding correlates or surrogates of protection and optimal timing of vaccination in pregnancy in the absence of correlating these with clinical efficacy in mothers and babies and using validated assays.

In this regard, speculative statements need to be removed accordingly and the descriptive nature of the data should be clarified.

Response:

Although we agree that there is no known absolute antibody titer or level that guarantees protection, data from large studies conducted across multiple sites and countries support that antibodies generated in response to SARS-CoV-2 infection¹⁻³ and vaccination⁴⁻⁶ do correlate with protection against severe disease in the adult population. With the recent CDC study demonstrating that maternal vaccination in pregnancy protects the neonate from hospitalization from COVID-19 in the first 6 months of life,⁷ our recent paper in JAMA showing persistence of maternal antibody up to 6 months of age in infants suggests that antibodies are mediating the protection at least in this particular circumstance,⁸ given that only maternal IgG is transferred to the neonate transplacentally, and neonates do not have their own T and B cell responses themselves as they were not vaccinated.

In light of the reviewer's point that our study was not designed to specifically assess clinical effectiveness outcomes, and thus we cannot know whether the differences we have observed in immunological responses in our cohort directly translate to clinical differences in protection, we have reworded the last sentence of the abstract to the following: "These results provide novel insight into the impact of platform and trimester of vaccination on maternal humoral immune response and transplacental antibody transfer." In the discussion, we have made substantive edits to sentences in which we discuss the implications of our findings with regards to maternal or neonatal protection (p.17, p.18) and we have reorganized the paragraph in which we discuss our findings in the context of what is known about the protection of COVID-19 vaccines during pregnancy, and to frame our results in the context of current vaccine recommendations (p. 20). We have added language to clarify that our study is descriptive in nature and does not directly correlate serological data with clinical vaccine effectiveness outcomes (p.19). We have also removed a sentence from the concluding paragraph of the Discussion in accordance with this Reviewer's suggestions.

Relevant references:

Sadarangani M, Nature Reviews 2021

Plotkin S, Clin Vacc Immunology 2010

Response: We have included these relevant references in the manuscript text (ref 90 and 91).

References:

1. Zohar T, Loos C, Fischinger S, et al. Compromised Humoral Functional Evolution Tracks with SARS-CoV-2 Mortality. *Cell*. 2020;183(6):1508-1519.e12.
2. Finch E, Lowe R, Fischinger S, et al. SARS-CoV-2 antibodies protect against reinfection for at least 6 months in a multicentre seroepidemiological workplace cohort. *PLoS Biol*. 2022;20(2):e3001531.
3. Harvey RA, Rassen JA, Kabelac CA, et al. Association of SARS-CoV-2 Seropositive Antibody Test With Risk of Future Infection. *JAMA Intern Med*. 2021;181(5):672-679.
4. Wei J, Pouwels KB, Stoesser N, et al. Antibody responses and correlates of protection in the general population after two doses of the ChAdOx1 or BNT162b2 vaccines. *Nat Med*. Published online February 14, 2022. doi:10.1038/s41591-022-01721-6
5. Amirthalingam G, Bernal JL, Andrews NJ, et al. Serological responses and vaccine effectiveness for extended COVID-19 vaccine schedules in England. *Nat Commun*. 2021;12(1):7217.
6. Gilbert PB, Montefiori DC, McDermott AB, et al. Immune correlates analysis of the mRNA-1273 COVID-19 vaccine efficacy clinical trial. *Science*. 2022;375(6576):43-50.
7. Halasa NB, Olson SM, Staat MA, et al. Effectiveness of Maternal Vaccination with mRNA COVID-19 Vaccine During Pregnancy Against COVID-19-Associated Hospitalization in Infants Aged <6 Months - 17 States, July 2021-January 2022. *MMWR Morb Mortal Wkly Rep*. 2022;71(7):264-270.
8. Shook LL, Atyeo CG, Yonker LM, et al. Durability of Anti-Spike Antibodies in Infants After Maternal COVID-19 Vaccination or Natural Infection. *JAMA*. Published online February 7, 2022. doi:10.1001/jama.2022.1206